



# Dry versus wet marine particle optical properties: RH dependence of depolarization ratio, backscatter and extinction from multiwavelength lidar measurements during SALTRACE

Moritz Haarig[1], Albert Ansmann[1], Josef Gasteiger[2], Konrad Kandler[3], Dietrich Althausen[1], Holger Baars[1], and David A. Farrell[4]

[1]Leibniz Institute for Tropospheric Research (TROPOS), Leipzig, Germany
[2]Universität Wien, Vienna, Austria
[3]Technische Universität Darmstadt, Darmstadt, Germany
[4]Caribbean Institute for Meteorology and Hydrology, Bridgetown, Barbados

*Correspondence to:* Moritz Haarig (haarig@tropos.de)

**Abstract.** Three-wavelength lidar observations of the depolarization ratio and the backscatter coefficient of marine aerosol as a function of relative humidity (RH) are presented. The humidity dependence of the extinction coefficient and the shape dependence of the lidar ratio were observed in the UV and visible. The phase transition from spherical sea salt particles to cubic-like sea salt crystals was observed under atmospheric conditions with a polarization lidar. The measurements were performed at Barbados (13°N, 59°W) during the SALTRACE winter campaign in February 2014. The radiosonde and Raman lidar observations show a drop in relative humidity below 50% in the marine aerosol layer simultaneously with a strong increase in particle linear depolarization ratio. Enhanced depolarization ratios (with systematic uncertainty) up to 0.12±0.08 (at 355 nm), 0.15±0.03 (at 532 nm) and 0.10±0.01 (at 1064 nm) were observed and compared to modeling studies of cubic sea salt particles. Lidar-derived correlations of the backscatter coefficient and the depolarization ratio with the relative humidity are reported with a 5-min time resolution. The scattering enhancement due to hygroscopic growth of the marine aerosol particles under atmospheric conditions was determined. Extinction enhancement factors from 40% to 80% RH of 2.05±0.82 at 355 nm, 3.73±0.86 at 532 nm and 5.37±1.25 at 1064 nm were found.

## 1 Introduction

Since more than 70% of the Earth are covered with water, the optical properties of marine particles must be carefully considered in radiative transfer schemes in global atmospheric models. This includes marine conditions with RH < 50% so that marine particles get dry, change their shape and thus their optical properties as we will demonstrate in this paper. The shape of sea salt particles strongly depends on the relative humidity (RH). At typical values of RH of > 80% in the marine boundary layer, the sea salt particles are liquid solution drops and thus spherical in shape. When RH decreases below 45%, they crystallize and become mostly cubic-like in shape (Tang et al., 1997). The change in shape leads to different optical properties, especially to changes in the linear depolarization ratio. Spheres have a linear depolarization ratio of ideally zero, non-spherical particles exhibit higher values (e.g. Gasteiger et al. (2011)). The different optical properties of dry and humid sea salt have to



be considered in various applications.

Satellite passive remote sensing as well as ground-based passive remote sensing (AERONET, e.g. Smirnov et al. (2002); Sayer et al. (2012) for marine environments) may be sometimes significantly affected by dry marine particles in marine environments (coastal regions during sea breeze effects). Cubic-like particles have a different scattering phase function than spherical

particles. Analogous to the mixture of Saharan dust (assume to be spheroidal in shape) and spherical anthropogenic particles, in the case of marine particles one would need an analysis scheme which considers cubic particles (and related scattering phase function) besides the spherical ones. The same should be considered in lidar inversion methods (e.g. Veselovskii et al. (2010); Müller et al. (2013)), when inverting microphysical properties over the oceans and coastal areas. Non-spherical particles can have a sensitive impact on their retrieval products, thus particle shape has to be carefully considered.

Aerosol classification from active remote sensing (Burton et al., 2012; Groß et al., 2013) based on the depolarization ratio will get trouble if dried marine aerosol with a high depolarization ratio is present. The polarization lidar observation used to separate dust from non-dust (e.g. Sugimoto and Lee (2006); Tesche et al. (2009); Mamouri and Ansmann (2014)) rely on the assumption that non-dust aerosol always produces depolarization ratios around 0.05 or less and significantly depolarization is only caused by dry irregularly shaped dust particles. The cubic-like sea salt particles will consequently be misinterpreted as

dust leading to an overestimation of the dust concentration.

Active remote sensing from satellites (CALIOP and EarthCARE) use the polarization technique to separate aerosol types. The derived optical properties like the extinction coefficient depend on the detection of the correct aerosol type to choose the appropriate lidar ratio (Omar et al., 2009). Dry sea salt particles may not be detected as sea salt, but misinterpreted as a different aerosol type with a higher lidar ratio, which leads to an overestimation of the extinction coefficient.

The change in shape characteristics of marine aerosol can be easily observed with polarization lidar (Murayama et al., 1999; Sakai et al., 2000). The potential of using Raman lidar to study the hygroscopic growth of aerosol particles was demonstrated by Veselovskii et al. (2009) for summer haze at the East coast of the United States. Granados-Muñoz et al. (2015) observed the decrease of depolarization ratio of marine aerosol mixtures with increasing RH over Granada, Southern Spain. The hygroscopic growth and the resulting scattering enhancement of marine aerosol was intensively studied in-situ at ground-level by artificially

humidification (Carrico et al., 2003; Swietlicki et al., 2008; Zieger et al., 2013).

This study aims to show how spherical and cubic-like sea salt particles can be separated by remote measurements with lidar and how these dried sea salt particles affect aerosol classification. When equipped with water vapor and nitrogen Raman channels the lidar delivers profiles of specific humidity. Combined with regularly available temperature profiles from radiosondes or models we even have RH together with the depolarization observations and can thus carefully measure changing particle shape

effects with RH, as a function of height. Sea salt crystallization and deliquescence was observed with a triple-wavelength polarization lidar (Haarig et al., 2017) in the marine boundary layer over the remote tropical Atlantic in the absence of any disturbing anthropogenic impact and lofted Saharan dust layer. In this way lidar can provide valuable information on the state of the marine aerosol layer from the point of view of optical properties.

By performing triple wavelength polarization lidar observations we provide combined information on the shape/size charac-

teristics of marine aerosol ensembles for the modeling community dealing with the optical properties of irregularly shaped



mineral dust and sea salt particles (Gasteiger et al., 2011; David et al., 2013; Kemppinen et al., 2015). Besides the depolarization ratio of the particles, we deliver extinction-to-backscatter ratios (lidar ratios) in addition which are also sensitive to changing particle properties.

The relative humidity ranged from 40% till more than 80% in the marine aerosol layer. Scattering enhancement factors with relative humidity are measured under these atmospheric conditions. The enhanced backscatter coefficient depicts the hygroscopic growth of the particles in terms of changing optical properties. For pure marine aerosol the hygroscopic growth factors and thus the backscatter enhancement are larger compared to continental or anthropogenic influenced maritime measurements (Zieger et al., 2013).

At the beginning we will give an introduction to sea salt aerosol under dry and humid conditions and show examples of sea salt particles collected above Barbados. The lidar measurements in the framework of the Saharan Aerosol Long-range Transport and Aerosol Cloud interaction Experiment (SALTRACE) (Weinzierl et al., 2017) will be described briefly. Then we will present our observations with a special focus on the phase transition from spherical solution droplets to crystalline sea salt particles and the scattering enhancement factor. In the discussion we compare our results to model calculations.

## 2  Sea salt under dry and humid conditions

The oceans are the source of marine aerosol, which consists mainly of sea salt and organic compounds from the sea surface (Gantt and Meskhidze, 2013). Marine aerosol from sea spray acts as cloud condensation nuclei and ice nucleating particles over the ocean where other types of particles are rare (DeMott et al., 2016; Kristensen et al., 2016). Marine aerosols are found in a wide size range from some nanometer ($10^{-9}$ m) up to several micrometer ($10^{-5}$ m) (Bates et al., 1998; Dadashazar et al., 2017). The Aitken mode (diameter < 70-80 nm) is dominated by organic material emitted from the ocean. New particle formation dominates the accumulation mode (80–300 nm) and the larger particles are mostly sea salt (e.g. Wex et al. (2016)). The sea salt particles may contain some organics on its surface (Middlebrook et al., 1998; Facchini et al., 2008; Laskin et al., 2012; Tervahattu et al., 2002). Size-resolved studies of marine aerosol (up to 2.5 $\mu$m) have been performed on Barbados by Wex et al. (2016). They found that the sea spray mode (>300 nm) contributes 4–10% to the total number, but 90% to the total mass, and thus volume. The number size distribution is dominated by the Aitken mode. But the surface area and the volume size distribution are dominated by the sea spray mode with negligible contribution of the Aitken mode, resulting in a bimodal size distribution (Wex et al. (2016), and personal communication with H. Wex). For the radiation studies as for lidar measurements the surface area and its bimodal distribution is decisive. In this study, we will focus on sea salt and its different shapes, which mainly forms the coarse or sea spray mode. The accumulation mode (later on called fine mode) consists of sea salt particles and newly formed organic particles.

Marine aerosol contains sea salt and organic compounds. Sea salt is the dominant component of the coarse mode and thus dominates the optical properties. Therefore most of the following investigations concern the sea salt and its changes with relative humidity.



Crystalline sodium chloride has a cubic shape, sulfates form frequently needle-like shapes (e.g., Kandler et al. (2007)). With different compounds present in sea-salt, mixed particle geometries can occur (e.g., Wise et al. (2007)). With increasing relative humidity the hygroscopic material takes up water, deliquesces and forms a spherical solution droplet. In humid marine environments, sea salt particles are of spherical shape. Consequently sea salt exists in two shape modes, spherical and non-spherical

crystalline.

The shape and thus the depolarization ratio is not only dependent on the relative humidity, but also on the chemical composition, leading to different crystalline shapes. NaCl is the major component of sea salt, but other salts such as $Na_2SO_4$, $MgCl_2$ and $MgSO_4$ and eventually some organics are part of atmospheric sea salt (Tang et al., 1997). These components prevent the perfectly cubic shape of dried sea salt.

The deliquescence behavior of a sodium chloride particle is shown in Fig 1. The particle grows with increasing relative humidity. Once the deliquescence point (in this case at a relative humidity of approximately 75%) is reached, it turns into a droplet. The studies of Tang et al. (1997) found the deliquescence point for sea salt at 70–74%, depending on the composition of the sea salt. The particles keep the spherical shape until the relative humidity decreases to 45–48% (Tang et al., 1997). As a result of the hysteresis effect, sea salt particles may exist in both shape modes between approximately 50% and 70% relative humidity,

depending on their individual history. These conditions may occur quite often over the oceans and coastal areas, as illustrated for example by Kandler et al. (2011). The hysteresis behavior has been studied previously (Carrico et al., 2003; Wise et al., 2005).

Atmospheric samples of dry salt particles (Fig. 2) were collected during the SALTRACE-1 campaign (Barbados, summer 2013). The samples are taken in the dust layer (2–4 km asl.) to ensure that the particles have been dried in airborne state to

be representative for atmospheric aerosol, in contrast to wet collection followed by drying on a substrate, which might lead to substrate effects. The four dry marine particles shown in Fig. 2 were collected at relative humidity between 28% and 39%. The X-ray fluorescence spectroscopy (XRF) pictures reveal, that particle (b) with a shape closer to a cube has a negligible contribution of sulfate, whereas particle (d) is more spherical in shape and has a larger contribution of sulfate. Also, organics might be the reason for the more spherical shape. Particle (c) shows the outline of a droplet, so this one was probable still deliquesced

at the time of collection. Consequently, it shows a considerably different shape than the other particles. Overall we see that sea salt particles have a non-spherical shape that could be approximated by a cube for a relative humidity below 40%. But the shape is not perfectly cubic as for pure sodium chloride (see Fig. 1). The edges of the sea salt particles dried in the atmosphere are smoother. In the following we will call the shape of crystalline sea salt 'cubic-like' to separate it from the spherical sea salt droplets under humid conditions. Compared to model results of perfectly cubic particles or pure NaCl salt particles investigated

in the laboratory, we should measure lower depolarization ratios for dried marine aerosol in the atmosphere, because of the smoothed cubic-like shape.

In most of the cases sea salt particles are spherical, because of the humid marine conditions in which they are predominantly found. This results in a low particle linear depolarization ratio of 0.03±0.01 at 532 nm (Groß et al., 2013). The range of depolarization values given in the mentioned publication is $0.01 - 0.11$. This indicates that not all cases classified as marine aerosol

consisted of spherical sea salt particles.



Field studies on dried marine particles are very rare. In lidar field measurements, first evidence of an enhanced depolarization ratio was reported by Murayama et al. (1999) for dry marine particles based on measurements in Tokyo. A clear separation from a potential dust influence was not possible. They observed a peak in the particle linear depolarization ratio of 0.1 at 532 nm during sea breeze.

Sakai et al. (2000) observed with a Raman lidar the relative humidity and the depolarization ratio over Nagoya, Japan, from 1994–1997. They found 532 nm particle linear depolarization ratios between 0.1 and 0.2 for 25– 45% RH at heights from 2–4 km in the free troposphere. Backward trajectories indicated pure marine conditions. These values are in good agreement with our observations.

In a laboratory study, Sakai et al. (2010) measured the particle linear depolarization ratio (PLDR) of spherical sea salt particles
of 0.01 with a mode radius of 0.08 $\mu$m. For crystalline sea salt particles they found a PLDR of 0.08 with a mode radius of 0.19 $\mu$m. Pure crystalline NaCl has a significantly higher depolarization ratio (PLDR=0.21) than atmospheric sea salt (Sakai et al., 2010). In a laboratory study Järvinen et al. (2016) observed pure NaCl particles with a depolarization ratio of 0.25.

Additionally to the change in particle shape, the size of sea salt aerosols changes with relative humidity. Due to water uptake sea salt aerosols are much larger under humid conditions and smaller under dry conditions. The process is known as hygro-
scopic growth (e.g. Zieger et al. (2013); Skupin et al. (2016)). The change in optical properties like the backscatter coefficient with varying relative humidity can be measured by a Raman lidar.

Early discrete dipole approximation (DDA) modeling attempts of spherical and cubic sea salt particles have been done by Murayama et al. (1999). For cubic particles larger than 0.8 $\mu$m a PLDR of 0.08-0.22 at 532 nm was predicted. David et al. (2013) used a T-matrix approach for cubic sea salt particles to model the depolarization ratio (approx. 0.16 in the visible and
UV) and the lidar ratio (19 sr in UV and 20 sr in the visible). The DDA approach for cubic particles including surface roughness (Kemppinen et al., 2015) leads to a PLDR 0.1–0.2 and a lidar ratio of 15–20 sr for the particle radius equal to the wavelength (size parameter = 6). Our observations presented in this paper can be explained by modeling the optical properties of cubic sea salt particles.

# 3 Methods

## 3.1 The SALTRACE project

The three SALTRACE field campaigns in 2013 and 2014 are the final observational efforts of the long-term SAMUM-SALTRACE project (Heintzenberg, 2009; Ansmann et al., 2011; Weinzierl et al., 2017). During SALTRACE, we investigated the Saharan dust properties after an atmospheric travel over 5–15 days and 5000–8000 km (Weinzierl et al., 2017; Haarig et al., 2016a, 2017). In the summer seasons of 2013 and 2014 (SALTRACE-1 and SALTRACE-3 in June-July), aged dust layers were
observed. To investigate aged mixtures of dust and biomass burning smoke after long-range transport, we performed an additional campaign in February-March 2014 (SALTRACE-2, winter transport regime). In February 2014 there was a period without aerosol transport from Africa, resulting in very clean marine conditions over Barbados. The SALTRACE lidar activities were complemented by shipborne lidar observations along the main Sahara dust transport route over the tropical North





Atlantic in April-May 2013 (Kanitz et al., 2013; Rittmeister et al., 2017).

The ground-based remote sensing station was deployed at the Caribbean Institute of Meteorology and Hydrology (CIMH) in Husbands, north of the capital Bridgetown at the west coast of Barbados (13.15°N, 59.62°W). The BERTHA lidar system (Backscatter Extinction lidar Ratio Temperature Humidity profiling Apparatus), an AERONET sun photometer (see AERONET web page, Barbados-SALTRACE site), and a Vaisala radiosonde station (RS92 for profiling of pressure, temperature, relative humidity, and the vector of the horizontal wind component) were operated at the field site. A second AERONET station (Ragged Point) is located at the east coast of Barbados, approximately 20 km away from the CIMH. This station is in use since 2007.

## 3.2 Three-wavelength lidar BERTHA

The multi-wavelength polarization Raman lidar BERTHA of the Leibniz Institute for Tropospheric Research (TROPOS) is a container-based mobile lidar system. As a unique feature, it enables the measurement of the depolarization ratio at three wavelengths simultaneously. A more detailed description of the lidar system and the polarization characteristics can be found in Haarig et al. (2017). Currently it operates as a 3+2+3 lidar system (3 backscatter coefficients, 2 extinction coefficients and 3 depolarization ratios) with an additional water vapor channel (407 nm) and a high spectral resolution channel at 532 nm. It has been used in a 3+3+2 configuration in Haarig et al. (2016b). The signals are detected with a range resolution of 7.5 m and a time resolution of 10 s.

The particle backscatter coefficient gives information about the particle layers. For particles with sizes comparable to the wavelength or larger, it is in first approximation proportional to the surface area of the bulk of particles. The extinction coefficient is determined from the transmission of the laser beam through the atmosphere. Both are calculated independently from the lidar signals via the Raman method (Ansmann et al., 1992). The backscatter-to-extinction ratio, also called lidar ratio, contains information about the particle size and shape, as well as about the refractive index. Therefore it is used together with the particle depolarization ratio to classify aerosol types (Burton et al., 2012; Groß et al., 2013). The 532 nm channels do not need an overlap correction, but the 355 nm channels has to be overlap corrected according to Wandinger and Ansmann (2002). The uncertainty especially of the extinction and the lidar ratio is therefore larger in the UV.

The particle linear depolarization ratio (PLDR) is a measure of the depolarization caused by the scattering of linear polarized light (defined as parallel) at atmospheric particles. It is defined as the ratio of cross polarized to parallel polarized light scattered back from aerosol particles. As an intensive parameter it is characteristic for a certain aerosol type. For spherical particles (droplets, wet marine particles) the PLDR is <0.03, whereas non-spherical particles have a higher PLDR (dust approx. 0.3 (at 532 nm), ice crystals approx. 0.5) (Groß et al., 2013). To ensure the good quality of the depolarization measurements, a $\Delta 90°$ calibration (Freudenthaler et al., 2009) was performed for each measurement. In the UV the systematic uncertainties are quite high. The calculation of the depolarization ratio follows Freudenthaler (2016) and is described in Haarig et al. (2017), where a detailed error estimation can be found.

For the calculation of the relative humidity, the temperature profile of the radiosonde is used. BERHTA measures the pure rotational Raman signals from nitrogen and oxygen from the 532-nm-emission wavelength to retrieve the temperature profile,



but the uncertainty is too large to retrieve the relative humidity with a reasonable uncertainty (Mattis et al., 2002). During the SALTRACE campaign, a Vaisalla RS92 radiosonde was launched for each measurement. The water vapor mixing ratio of the radiosonde is used to calibrate the water vapor mixing ratio derived by the ratio of the hydrogen (407 nm) and nitrogen (387 nm) Raman signal of the lidar (Whiteman et al., 1992). Due to the weak 407 nm signal, the technique can be used at

nighttime only. Using the the water vapor mixing ratio profiles of the lidar and the temperature and pressure profile of the radiosonde the temporal and vertical evolution of relative humidity can be derived. This can be seen in the lower panel of Fig. 4 and 5. The relative error of the water vapor mixing ratio caused by calibration and signal noise was < 5% at all height up to MAL top. The temperature of the radiosonde is used for the two hours of measurement, so an uncertainty of 1 K is reasonable. These errors lead to a maximal relative uncertainty in the relative humidity of 12%, that signifies for the dry relative humidity

40±5%. A detailed error estimation for the relative humidity derived with a Raman lidar can be found in Mattis et al. (2002).

### 3.3    The DDA model for cubic sodium chloride

We simulate optical properties of dry sea salt particles as cubes with the refractive index $m$ of sodium chloride, provided by Eldridge and Palik (1985), i.e. with values $m$=1.582 at $\lambda$=355 nm, $m$=1.549 at $\lambda$=532 nm, and $m$=1.531 at $\lambda$=1064 nm.
A range of volume-equivalent particle radii up to 2 $\mu$m is covered by modeling with the Discrete Dipole Approximation

code ADDA (Yurkin and Hoekstra, 2011) with logarithmically equidistant size steps of about a factor of 1.1. We use the DDA formulation 'filtered coupled dipoles' (Piller and Martin, 1998) included in ADDA, which was also used for example by Gasteiger et al. (2011), and use 8 dipoles (dpl) per wavelength. To simulate random particle orientation, DDA runs for 100 orientations were carried out for each particle. The weighted distributions of particle orientations were selected according to those presented by Sloan and Womersley (2004). For each DDA run, the optical properties were averaged over 64 scattering

planes rotated around the incident light direction.
To estimate the accuracy of these model simulations, the scattering problems were modeled with settings associated with higher accuracy, i.e. one case with increased number of dipoles per wavelength (dpl=12 instead of 8) and another case where we increased the number of orientations from 100 to 225. The uncertainty is estimated for the lidar ratio of single randomly-oriented particles to be on the order of ±10% and for their linear depolarization ratios about ±0.02. A similar uncertainty is

estimated for the final optical properties of the bulk sea salt aerosol. The estimate for the bulk properties is based on considering that on the one hand the uncertainty is reduced by averaging over size but on the other hand also the applied size distribution, particle shape, and refractive index are connected with uncertainties.

### 4    Observations

The island of Barbados is ideal to observe pure marine conditions. It is the eastern-most island of the Caribbean and located

in the trade wind zone with predominant wind direction from the east. In winter, the inner tropical convergence zone is shifted to the southern hemisphere, so the air masses originating from the African continent are transported to South America (e.g. Baars et al. (2011)), leaving the Caribbean under marine influence.



During the SALTRACE-2 campaign (16 February – 8 March 2014), a layer of enhanced cross polarized signal was observed for several days. Figure 3 shows the most pronounced days (23 – 25 February 2014), but similar features were observed for most of the clean marine days in February 2014. The marine aerosol extends up to the strong trade wind inversion at around 2 km height (Marine Aerosol Layer - MAL). The marine aerosol layer is defined by the predominance of marine aerosol. It is

on on top of the marine boundary layer (MBL), which is the well mixed boundary layer over the ocean reaching up to 0.7–1 km. In the free troposphere above the MAL, there is no dust or other continental aerosol, that could interfere by downward mixing with the marine aerosol as it is the case in other seasons (Groß et al., 2016).

The two night measurements of 23 and 24 February 2014 will be discussed in the following to demonstrate how the decrease in RH leads to higher depolarization ratios due to changes in the shape of the sea salt particles.

Firstly the observations during the two night measurements will be described. Then the changes in shape and size with relative humidity will be discussed using the particle linear depolarization ratio and the particle backscatter coefficient, respectively.

## 4.1 Profile measurements of backscatter, depolarization and relative humidity

An overview of the two case studies discussed in this publication is given in Fig 4 and 5.

On the 23 February 2014 a complex scenario was measured. The marine aerosol layer was not humid up to the trade wind

inversion (at 2000 m), but the relative humidity decreased steadily from 80% at 250 m height to 35% at 1000 m height. The slow decrease is ideal to study the drying process of marine aerosol particles under atmospheric conditions. Above the relative humidity increased with height up to 80% at 1800 m. Then a fast decrease of the relative humidity (80% to less than 10%) was found at the trade wind inversion between 1850 and 2150 m. This feature was observed for most of the measurements under clean marine conditions in February 2014. The time-height display of the relative humidity is shown in the lower panel

of Fig. 4. The increased signal in the cross polarized channel and the volume depolarization ratio at 1064 nm are shown in the upper panels of Fig. 4. In parts with low relative humidity, mostly between 1000 and 1600 m, the volume depolarization ratio is high indicating non-spherical particles.

On the 24 February 2014 only a thin layer of dried marine particles was observed. The humid marine aerosol layer reached up to 2 km (Fig. 5). The feature of interest is the enhanced 1064 nm cross polarized signal in the upper 200 m of the aerosol layer.

The radiosonde launched at 23:07 UTC (19:07 local time (LT)) shows a strong temperature inversion of 4 K at 2000 m height (Fig. 9). In the marine aerosol layer, RH was about 65% to 80%. RH decreased to values of about 5% within 200 m at MAL top. The relative humidity (Fig. 5 bottom) indicates that the scenario does not change for the two hours average (18:11-20:20 LT). Marine particles lost their water shell and their spherical shape at the top of the marine aerosol layer. This caused an enhanced depolarization ratio, as can be seen in the central panel of Fig. 5.

For the further discussion it is important to show, that only marine aerosol was present over Barbados, forming the marine aerosol layer. HYSPLIT backward trajectories (Stein et al., 2015; HYSPLIT, 2017) and AERONET (AERONET, 2017) observations are used to demonstrate the presence of only marine aerosol. The ensemble of 7-days backward trajectories for 23 February (Fig. 6 left) indicate marine sources over the Atlantic for the air mass, with only a slight possibility of influences from West Sahara, where the air masses could have passed above 2 km height. At the 24 February the ensemble of 7-days



backward trajectories shows the entrainment of air masses from the free troposphere at 2000 m height. The air masses from the free troposphere did not have any contact to the ground in the last 7 days, so there is no other source of aerosols. The contact of dry air masses from the free troposphere with the humid marine air masses over the Western Atlantic result in a drying process of the marine aerosols as observed by the lidar measurements. Considering the ensembles of trajectories at different heights

below 2000 m for both nights (not shown), none of the 10-days back trajectory passed anything else but the ocean at heights below 2 km, where aerosol load could have taken place. This is a clear indication for the marine origin of the aerosols in the MAL at that two case studies presented in this publication.

The measurements of the AERONET (AERONET, 2017) sun photometers at Ragged Point and at the field site (Barbados-SALTRACE) are shown in Fig. 7. The optical thickness (AOT at 500 nm) is low (AOT500 < 0.1), that is typical for marine

environments. Sayer et al. (2012) used AOT500 ≤ 0.2 to constrain marine conditions from AERONET. For the Ångström exponent (AE at 440 − 870 nm), they use 0.1 ≤ AE ≤ 1. In our case the Ångström exponent is lower (-0.2 < AE < 0.2), indicating larger particles.

Overall we are very confident, that only pure marine aerosol was present in our measurements and no other aerosol type interfered with our measurements. Air mass back trajectories and column integrated quantities indicate clean marine conditions.

The vertical profiles of the particle backscatter coefficient and the PLDR for the three wavelengths (355, 532 and 1064 nm) of the BERTHA lidar system are shown in Fig. 8 and 9. The radiosonde launched at the field site provides profiles of temperature, potential temperature and relative humidity. A good agreement is achieved between the radiosonde relative humidity and the temporally averaged relative humidity profile retrieved from the lidar. The profile of the relative humidity from the radiosonde was added to the profiles of the backscatter coefficient and the PLDR to show the different layers with respect to RH. The crys-

tallization point for sea salt of 45-48% RH (Tang et al., 1997) is marked with a dashed line. The depolarization ratio increases strongly when RH drops below this point. The change in the backscatter coefficient is less pronounced, but it decreases slightly when RH drops below 48%.

The profiles for the 23 February 2014 (Fig. 8) are the average of 30 min (23:45-00:15 UTC) around the launch of the radiosonde at 00:00 UTC.When the relative humidity decreases below the crystallization point at 810 m height, the particle linear depo-

larization ratio increases up to 0.153±0.035 (at 532 nm) at around 1150 m height. When the relative humidity increases again, it reaches the deliquescence point (70–74% RH) at 1780 m and the PLDR decreases below 0.02 (532 nm). This behavior will be discussed with a higher temporal resolution in the next subsection. The second decrease in relative humidity at the trade wind inversion leads to a less pronounced peak in the PLDR (value ± systematic uncertainty) of 0.069±0.161 at 355 nm, 0.079±0.036 at 532 nm, and 0.063±0.018 at 1064 nm around 2000 m.

In the 2-hour mean profiles of the 24 February 2014 (Fig. 9), a maximum PLDR of 0.055±0.109 (355 nm), 0.068±0.035 (532 nm) and 0.038±0.010 (1064 nm) is reached in the thin layer of dried marine aerosol at MAL top (100 m vertical smoothing). In the humid marine aerosol layer below, the PLDR is below 0.02. The profile of the relative humidity shows the coincidence of the decrease of the relative humidity with the increase of PLDR.

The Raman lidar method allows us to derive the extinction coefficient independently of the backscatter coefficient and therefor

to measure the lidar ratio (extinction-to-backscatter ratio), shown in Fig. 10. In the humid marine aerosol layer on 24 February




the lidar ratio is 21±5 sr (at 355 nm) and 23±2 sr (at 532 nm), which is typical for marine environments (Müller et al., 2007; Groß et al., 2013). The layer of dried marine aerosol at MAL top was too thin to derive a lidar ratio. For the thicker dried marine aerosol layer on 23 February (1000-1600 m) a lidar ratio can be derived in the center of the layer (1100-1300 m) to avoid smoothing effects. We measured a lidar ratio of 28±6 sr (at 355 nm) and 25±3 sr (at 532 nm) for cubic-like sea salt

particles. The results are summarized in Tab 1 for dry and humid conditions. There is a slight increase in lidar ratio for dried sea salt compared to humid sea salt. But the change is not significant and has to be proven in further measurements. The lidar ratio of cubic-like sea salt particles has not been measured yet. David et al. (2013) simulated with the T-matrix approach a lidar ratio of 19 sr and 20 sr at 355 nm and 532 nm for cubic sea salt particles.

### 4.2 Observation of the phase transition in the depolarization ratio

The phase transition between spherical sea salt droplets and cubic-like sea salt crystals was described in section 2, where the relevant literature is given. Using a polarization lidar offers the opportunity to study the transition in particle shape from spherical (low depolarization) to non-spherical, e.g. cubic-like (higher depolarization) under atmospheric conditions. Using a Raman lidar with a water vapor channel (407 nm) has the advantage to provide the relative humidity with a higher temporal and vertical resolution (5 min, 50 m) compared to just one radiosonde per measurement. The unique three-wavelength polarization

Raman lidar BERTHA enables the measurement of the PLDR in the ultra violet (355 nm), the visible (532 nm) and the near infrared (1064 nm) simultaneously with the relative humidity.

The drying process may occur within the marine aerosol layer or on top of it, where dry air from the free troposphere is mixed in. The dry marine layer was often observed on MAL top under clean marine conditions as has been shown for 23 and 24 February 2014. There the increase in depolarization is less pronounced as in the case, where the dried marine aerosol was

found within the MAL (24 Feb 2014). The reason might be that both shape modes (spherical and cubic-like) exist at the same time in the thin layer on MAL top, because the of the fast decrease in RH as it occurs at a strong temperature inversion. The vertical resolution of the data (50 m) in a 200 m deep layer might also be a reason for lower measured depolarization ratios. But still depolarization ratios up to 0.079±0.036 (at 532 nm, ± sys. uncertainty) are measured, indicating a phase transition from spherical sea salt droplets to crystalline sea salt, in line with laboratory measurements of 0.08 at 532 nm (Sakai et al.,

25 2010).

The correlation between the PLDR and the RH for the 23 February 2014 is shown in the lower panel of Fig. 11. Only the decrease in relative humidity is depicted here (375 m till 1100 m height). We observe the drying and not the humidification of aerosols. Therefore the crystallization point (45–48% RH) is more important than the deliquescence point (70–74% RH). From 80% – 50% RH the PLDR increases slightly with decreasing RH, but remains ≤ 0.02 (VIS, NIR) and ≤0.03 (UV, due

to higher noise level). This observations are in line with typical depolarization values for marine aerosol as used for aerosol classification schemes (e.g. Groß et al. (2013)). The sea salt aerosol has a spherical shape under humid conditions. At around 50% RH the PLDR increases drastically indicating a significant change in particle shape from spherical to non-spherical (cubic-like) particles. The PLDR reaches maximum values (with sys. uncertainty) of about 0.12±0.08 (at 355 nm), 0.15±0.03 (at 532 nm) and 0.10±0.01 (at 1064 nm) at a relative humidity of around 40%. The driest parts in our measurements (35% RH)





lead to no more significant increase in PLDR. After the phase transition from spherical sea salt droplets to cubic-like sea salt crystals, the depolarization ratio stays at the high level found for cubic-like particles. A further drying was not observed yet, so we can only speculate about the depolarization ratio for sea salt particles under very dry conditions (0-35% RH).

Furthermore the measurement of the 23 February 2014 contains information about the humidification process. Between 1100 and 1800 m height RH is increasing again up to 80%. Figure 12 shows the PLDR versus RH up to 1800 m with red circles (<1100 m) and purple stars (>1100 m). The humidification process (purple stars) does not follow the same line as the drying process (red circles). The sea salt remains in its cubic-like shape causing an enhanced depolarization ratio, which slowly decrease to values below 0.02 typical for spherical sea salt particles (above around 65-70% RH). The hysteresis effect leads to the existence of both shape modes between 50% and 70% RH.

## 4.3 Scattering enhancement factors of pure marine aerosol

Beside the change in particles shape, the particle size is changing with decreasing RH. Sea salt particles grow with the water uptake due to increasing RH. The particle backscatter coefficient is proportional to the surface area of the scattering particles and therefore a good indication for particle growth (if the number concentration stays the same). For the further investigation the measurement of 23 February 2014 is used with 50 m vertical and 5 min temporal resolution. In the upper row of Fig. 11 the particle backscatter coefficient $\beta$ is plotted against the RH for the height interval 375-1100 m for the three wavelengths ($\lambda$= 355, 532, 1064 nm).

The backscatter enhancement factor $f_\beta(\mathrm{RH},\lambda)$ is calculated from these data:

$$f_\beta(RH,\lambda) = \frac{\beta(RH,\lambda)}{\beta(40\%,\lambda)} \tag{1}$$

Under atmospheric conditions it is hardly possible to get dry aerosols (RH<10%) in the marine aerosol layer. Therefore the backscatter coefficient at 40% RH was chosen as reference value to normalize the data, as it has been done in literature (Skupin et al., 2016). Lower values than 35% RH were not accessible under the measurement conditions over Barbados. The relative humidity of 40% is below the crystallization point, so sea salt particles should not shrink significantly at further decreasing RH. Other lidar-based atmospheric studies used 40% or 60% as reference relative humidity (Granados-Muñoz et al., 2015; Veselovskii et al., 2009) since it was the lowest value found in their measurements.

To parametrize the backscatter enhancement factor, a model derived by Hänel is used (Kasten, 1969; Hänel, 1976, 1984):

$$f_\beta(RH,\lambda) = A * (1 - RH/100)^{-\gamma} \tag{2}$$

The parameter $A$ gives the extrapolated value at 0% RH and the exponent $\gamma$ is related to the hygroscopicity. This parametrization (sometimes with $A$=1 for starting at 0% RH) has been used by various investigators (Carrico et al., 2003; Veselovskii et al., 2009; Skupin et al., 2016).

The backscatter enhancement factors are shown in Fig. 13 with the parametrization according to equation (2). The fit parameters are listed in Table 2. The relative humidity has fairly reached 80% in the used height interval. $f_\beta$(75–80%) is the averaged value between 75% and 80% RH. Additionally $f_\beta$(80%) was calculated extrapolating the fit. Both values of the backscatter



enhancement factor with respect to the reference relative humidity of 40% can be found in Tab. 2. The extrapolated values are very high and the measured values at 75–80% RH do not support these high enhancement factors. For the further discussion we will only use the measured enhancement factors. The error of the backscatter enhancement factors results from the standard deviation of the mean values at 40% and 75–80% RH.

5 For a better comparison to reported literature values, we convert our backscatter enhancement factors $f_\beta$ to extinction enhancement factors $f_\alpha$ by means of the backscatter-to-extinction ratio (lidar ratio $S$), which was measured for wet and dry marine particles and is listed in Tab 1.

$$f_\alpha(RH,\lambda) = \frac{S_\text{wet}}{S_\text{dry}} f_\beta(RH,\lambda) \tag{3}$$

The extinction enhancement factor does not depend on the 180° backscatter direction any more and is thus more universal. For 10 1064 nm the modeled lidar ratios have to be used. The errors of the lidar ratios are included in the error of $f_\alpha$.

The extinction enhancement factors of pure marine aerosol range from 2.05±0.82 (at 355 nm), to 3.73±0.86 (at 532 nm) and 5.37±1.25 (at 1064 nm). A clear wavelength dependence can be seen. The enhancement of the backscatter and extinction coefficient is stronger at larger wavelengths. Qualitatively the same wavelength dependence was observed by Kotchenruther et al. (1999) for the wavelengths 450, 550 and 700 nm, where the relative increase of the scattering enhancement factor from one 15 wavelength to the next is about 8%. Zieger et al. (2013) stated that the wavelength dependence (450, 550, 700 nm) for marine aerosol is small (<5%) in their observed wavelength range.

The scattering enhancement factors in the present study are in the upper range of reported literature values for marine aerosol. It might be due to less influence by other aerosol types on Barbados, where 5000 km upwind is only ocean. Zieger et al. (2013) reported mean scattering enhancement factors f(85%, 550 nm) for sea salt of 2.28 at Mace Head, Ireland, 2.86 at Ny Ålesund, 20 Svalbard, and 3.38 at Cabauw, Netherlands. Carrico et al. (2003) measured f(82%, 550 nm) = 2.45 (diameter < 10 $\mu$m) and 2.95 (diameter < 1 $\mu$m) for maritime air in the North Pacific Ocean. A different size distributions for marine aerosol over Barbados might also be a reason for higher scattering enhancement. In contrast to the optical properties like the scattering enhancement, some studies investigated the hygroscopic growth in terms of diameter. Swietlicki et al. (2008) gives a review over the small marine particles (diameter < 500 nm). For sea salt they found hygroscopic growth factors ranging between 1.77 and 2.14 for 25 90% RH.

It should be pointed out, that there is a difference between the hygroscopic growth factors (Swietlicki et al., 2008; Veselovskii et al., 2009) calculated with respect to particle diameter and scattering enhancement factors dealing with the resulting changes in optical properties (Zieger et al., 2013; Carrico et al., 2003). The backscatter coefficient observed in this study is proportional to the surface area of the particles. If the particle diameter changes by a factor of 2, the backscatter should change by a factor of 30 4.

## 5 Comparison with optical modeling of sea salt cubes

Cubic sodium chloride particles were modeled using the Discrete Dipole Approximation (DDA). The model settings are described in section 3.3. The lidar ratio and the PLDR for each wavelength were modeled particle size resolved (Fig. 14). If the





particle diameter is around the detecting wavelength and smaller, the lidar ratio is quite high (up to more than 100 sr). Above a diameter of twice the wavelength (radius = wavelength), the lidar ratio is 20±10 sr with higher values for larger wavelengths. If the particle diameter is smaller than the detecting wavelength, the PLDR is very small. A significant depolarization is produced for particle diameters equal or larger than the detecting wavelength. Then values up to 0.27 are reached strongly depending

on particle size. In the atmosphere we have always a distribution of particles sizes. Measuring the PLDR at three wavelengths gives us additional information about the particles sizes.

We consider the size distribution inverted by the AERONET algorithm from measurements at Ragged Point on 23 February 2014 at 12:31 UTC (Fig. 15). It is version 1.5 only, but the inversion of that measurement resulted in the lowest residual error of the sky radiance on that day. We assume that the optical depth during that measurement was dominated by wet marine

particles. To calculate the optical properties for dry marine particles we assume that their size is halved compared to the size obtained by AERONET. The results are compared to the measurements in Table 1. In the UV and visible the simulated PLDR agrees with the measurements. Using the halved radius improves the agreement. For the near infrared the model overestimates the measured value.

The T-matrix results for cubic sea salt (David et al., 2013) are given for comparison (Tab 1). They agree quite well with the

maximum particle linear depolarization ratios measured in the UV and visible.

The modeled lidar ratios are smaller than the measured ones. The uncertainty for dried marine lidar ratio especially in the UV is high. Overall there is no large difference between wet and dry marine lidar ratios compared to other aerosol types (mineral dust 55±7 sr, smoke 79±17 sr at 532 nm, Tesche et al. (2011)).

## 6   Conclusions

The phase transition between spherical sea salt droplets and cubic-like sea salt crystals has been observed under pure marine conditions over Barbados. The particle linear depolarization ratio measured with a Raman lidar at three wavelengths is enhanced when the relative humidity drops below 50%. The combination of polarization and water vapor measurements with lidar offers the opportunity to study this behavior vertically resolved.

A layer of dried marine aerosol is often observed on top of the marine aerosol layer through the enhanced depolarization ratio

as has been shown in the case of 23 and 24 February 2014 over Barbados. We suppose that this phenomena occurs quite frequently over the ocean, when the humid air masses over the water get into contact with dry air masses above. It might occur at any marine location and is transported over the continent, but it is not easy to detect the dried marine aerosol unambiguously if there are different aerosol sources around. Further evidence of dried marine particles in the atmosphere were found in lidar observations over the Southern Atlantic aboard a research vessel 'Polarstern' (Bohlmann, 2017) and in our recent lidar observa-

tions in the Eastern Mediterranean at Limassol, Cyprus and at Haifa, Israel. Satellite based studies, for example with CALIPSO or EarthCARE, would be helpful to assess the global occurrence of dried marine aerosol. The particle linear depolarization ratio in these dried marine layers is enhanced (0.05–0.07 at 355 nm; 0.07–0.08 at 532 nm and 0.04–0.06 at 1064 nm), but lower than in the pronounced dry layer within the MAL measured on 23 February 2014.





The change in particle shape and thus its radiative properties should be carefully considered in radiative transfer studies and (lidar based) aerosol classification schemes. Most aerosol classification schemes (Burton et al., 2012; Groß et al., 2013; Mamouri and Ansmann, 2014) use the particle depolarization ratio as a basic criteria for certain aerosol types. Dried marine aerosol has a significant higher depolarization ratio (up to 0.15 at 532 nm) compared to wet marine (<0.05), which is usually

assumed for aerosol types like 'marine'. A layer of cubic-like marine aerosol on top of the marine aerosol layer would change the radiative transfer over the ocean and thus the global radiative transfer as the oceans cover a large part of the Earth. Inversion algorithms as used in AERONET may be affected as well. They separate between spheres and spheroids (Dubovik et al., 2006). For dry marine cases, we would suggest, that a cubic model could be included. Dry marine cases (RH <50%) were not observed very often with AERONET. Sayer et al. (2012) found 4 cases with near-surface RH <50% and 67 with RH 50–60%

out of more than 2500 marine measurements. But as we have shown most of the dried marine aerosol was found in higher altitudes.

We compared the optical properties at three wavelengths of dry and wet marine particles with relative humidity of 40% and 80%, respectively. The backscatter coefficient of wet marine particles is 2.73±0.72 (at 355 nm), 4.05±0.72 (at 532 nm) and 5.54±0.97 (at 1064 nm) times larger than for dry marine particles. The extinction enhancement factor for the same RH range is

2.05±0.82 (at 355 nm), 3.73±0.86 (at 532 nm) and 5.37±1.25 (at 1064 nm). The depolarization ratio for dry marine particles (40% RH) is on average (with standard deviation) 0.11±0.03 (at 355 nm), 0.11±0.03 (at 532 nm) and 0.09±0.01 (at 1064 nm) with maximum values reaching (with systematic uncertainty) 0.12±0.08 (at 355 nm), 0.15±0.03 (at 532 nm) and 0.10±0.01 at 1064 nm. These results are given in Tab 1 and 2.

The existence of cubic-like and spherical salt particles is known since a long time, but this study wants to point out the at-

mospheric relevance. Cubic-like sea salt has been measured under atmospheric conditions. The two shape modes of sea salt (spherical and cubic-like) exist under atmospheric conditions over the ocean and should be considered in future aerosol studies.

*Acknowledgements.* K. Kandler acknowledges support from the Deutsche Forschungsgemeinschaft (grant KA 2280/2).



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



**Table 1.** Measurement and simulation of the lidar ratio and the particle linear depolarization ratio for wet and dry marine particles. The depolarization ratio is measured at the indicated relative humidity (5 min temporal, 50 m vertical resolution). The lidar ratio is is averaged in the indicated time interval in the center of the layer (profiles in Fig 10). The lidar ratio at 1064 nm could not be measured in this configuration of the lidar system (see Haarig et al. (2016b)). The mean value and its standard deviation is given. For the DDA simulation of spherical and cubic particles the AERONET size distribution (SD) from Ragged Point at 12:31 UTC on 23 February 2014 is taken. In order to mimic the dried marine particle the radius of the same size distribution was divided in half (half radius). For the wet marine particles a mixture of 7:1 parts of water to salt was used. The model uncertainties are described with the model in section 3.3. The T-matrix results for a typical size distribution for sea salt (O'Dowd et al., 1997) are taken from David et al. (2013). The modeled uncertainties are extremely small, less than 1 for the lidar ratio.

| | **Lidar ratio** (sr) | | | **Particle depolarization ratio** (%) | | | Comments |
|---|---|---|---|---|---|---|---|
| Wavelength (nm) | 355 | 532 | 1064 | 355 | 532 | 1064 | |
| **Measurement** | | | | | | | |
| **Wet** (RH=80%) | $21 \pm 5$ | $23 \pm 2$ | – | $\leq 3.0$ | $\leq 2.0$ | $\leq 2.0$ | 2014-02-24 22:11-00:21 UTC, <1800 m |
| **Dry** (RH=40%) | $28 \pm 6$ | $25 \pm 3$ | – | $10.7 \pm 3.1$ | $11.3 \pm 2.6$ | $8.7 \pm 1.2$ | 2014-02-23 23:38-01:08 UTC, 1100 m |
| **Simulation** | | | | | | | |
| **Wet** (Spherical) | $22 \pm 2$ | $27 \pm 3$ | $35 \pm 4$ | 0 | 0 | 0 | DDA, Aeronet SD 2014-02-23 |
| **Dry** (Cubic) | $10 \pm 1$ | $16 \pm 2$ | $29 \pm 3$ | $8.7 \pm 2.0$ | $11.7 \pm 2.0$ | $14.9 \pm 2.0$ | DDA, Aeronet SD 2014-02-23 |
| | $13 \pm 1$ | $19 \pm 2$ | $36 \pm 4$ | $10.3 \pm 2.0$ | $12.9 \pm 2.0$ | $12.1 \pm 2.0$ | DDA, Aeronet SD 2014-02-23, half radius |
| | 19 | 20 | – | $15.9 \pm 0.1$ | $16.2 \pm 0.1$ | – | T-Matrix, sea salt SD*, David et al. (2013) |

\* Size distribution from O'Dowd et al. (1997)

**Table 2.** The fit parameters for the backscatter enhancement factor according to equation (2). The backscatter enhancement factor $f_\beta(80\%)$ calculated with these fit parameters is compared to the measured factor $f_\beta(75-80\%)$ between 75% and 80% RH. There are too less values measured at 80% RH. The extinction enhancement factor $f_\alpha(75-80\% RH)$ was derived by equation (3) to compare with the literature.

| Wavelength | Fit | | | Measurement | |
|---|---|---|---|---|---|
| | A | $\gamma$ | $f_\beta(80\%)$ | $f_\beta(75-80\%)$ | $f_\alpha(75-80\%)$ |
| 355 nm | 0.532 | 1.077 | 3.01 | $2.73 \pm 0.72$ | $2.05 \pm 0.82$ |
| 532 nm | 0.529 | 1.487 | 5.79 | $4.05 \pm 0.72$ | $3.73 \pm 0.86$ |
| 1064 nm | 0.489 | 1.837 | 9.40 | $5.54 \pm 0.97$ | $5.37 \pm 1.25$* |

\* modeled lidar ratios used ($S_{wet}$=35 sr; $S_{dry}$=36 sr; Tab 1)



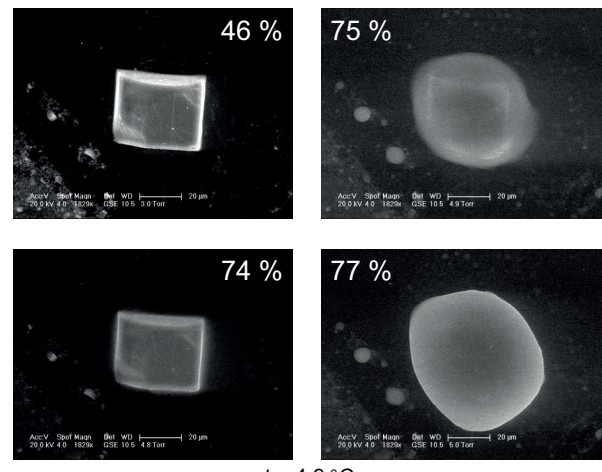

t = 4.9 °C

**Figure 1.** Sodium chloride deliquescence at a relative humidity (RH) of 75% observed at laboratory conditions (at 4.9°C). The dry cubic particle with sharp edges at RH of 46% becomes surrounded by a liquid sphere when RH is increased to 75%.





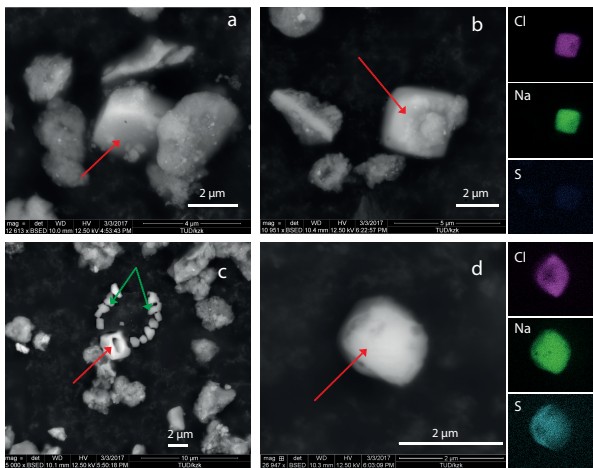

**Figure 2.** Images of dry atmospheric sea salt particles (red arrows) surrounded by Saharan dust particles. The particles were sampled aboard the Falcon research aircraft (Weinzierl et al., 2017) at different heights over Barbados on (a) 21 June, (b) 22 June, and (c,d) 23 June 2013 during SALTRACE-1. Sampling altitude and conditions are (a) 2560 m asl., 15°C, 29% RH; (b) 3550 m asl., 8°C, 28% RH; (c) 3570 m asl., 7°C, 39% RH; (d) 3230 m asl., 8°C, 34% RH. For the sea salt particles in (b) and (d) the XRF-Images are included showing that chloride (Cl) and sodium (Na) are the main components. The sulfate (S) component is negligible for particle (b) but significant for particle (d) which exhibits a more spherical shape. (c) shows an outline of a former droplet (green arrows), indicating a still (partial) deliquesced state during collection. The white bar in the bottom right corner indicates 2$\mu$m.



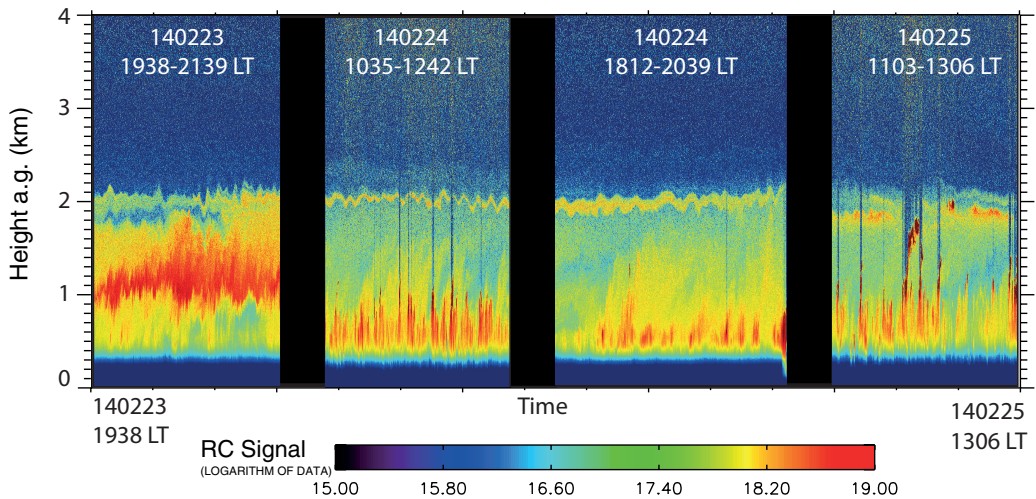

**Figure 3.** Three days of lidar observations (23–25 February 2014) of layers with dried marine particles in the marine aerosol layer (MAL). On 23 February (left panel), a vertically extended layer of dried sea salt particles (red area) occurred on top of the convective boundary layer. A continuously, 100-200 m thick layer with dried marine particles (yellow layer at MAL top) was present over the whole day on 24 February 2014, this layer was still present on 25 February. The convectively active marine boundary layer (MBL) reaches to about 1 km on all three days and permanently pushed marine aerosol upward. The shown range-corrected 1064 nm backscatter signal (cross-polarized channel) is most sensitive to enhanced light depolarization produced by dry marine aerosol. LT: local time





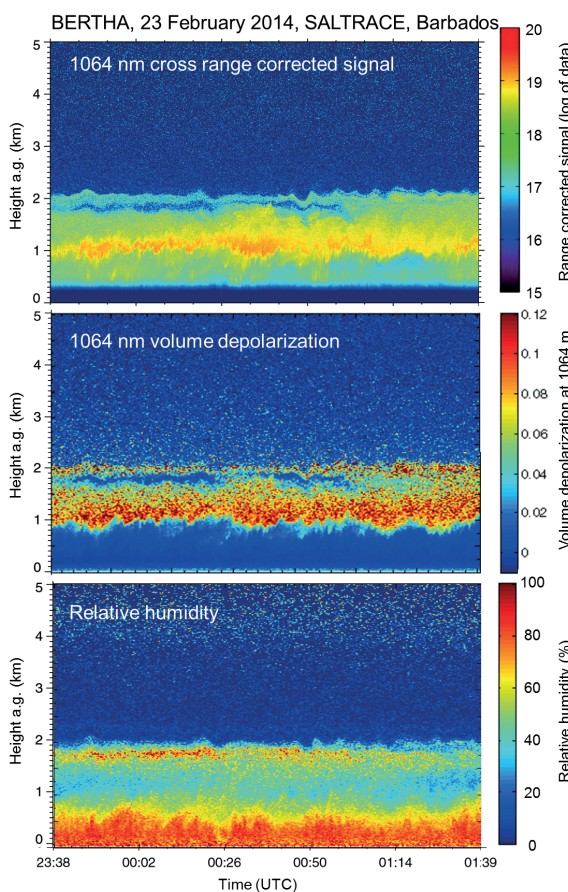

**Figure 4.** Marine aerosol layer (MAL) over Barbados on 23 Februray 2014, 19:38-21:39 local time with the cloudfree MBL reaching to 0.8-1 km height (indicated by a low depolarization ratio and high relative humidity) and an extended layer between about 1 and 2.1 km height with dried marine sea salt particles causing enhanced light depolarization (central panel, yellow-red areas) at low relative humidity of <50% (bottom panel, bluish areas). The cross-polarized 1064 nm signal (top panel) highlights the layer with dried marine particles (yellow areas).




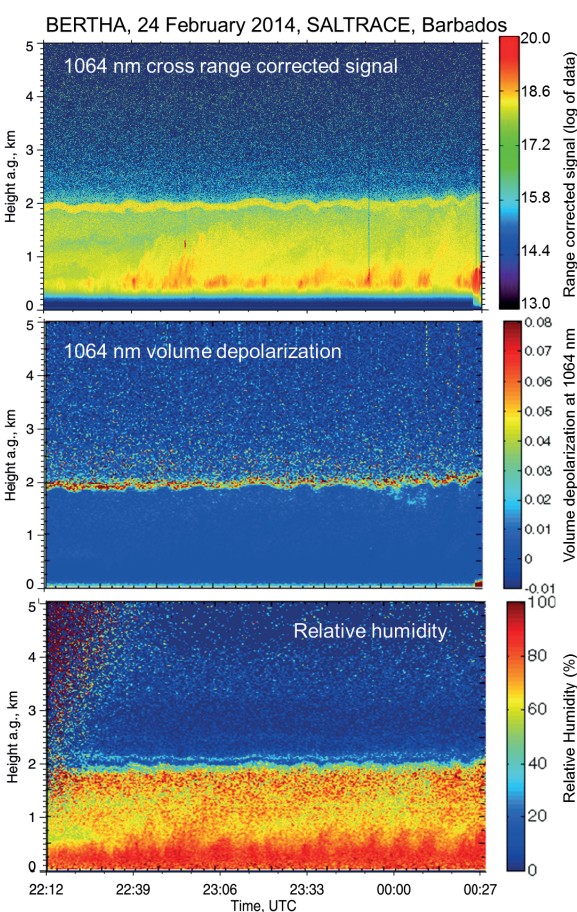

**Figure 5.** Same as Fig. 4, except for 24 February 2014, 18:12-20:27 local time. The MAL top is again close to 2 km height. The convective MBL reaches to about 800 m height and permanently pushes sea salt particles into the upper part of the MAL. RH is high throughout the MAL (and thus depolarization ratio caused by spherical, wet marine particles is low). Only at MAL top, dried marine particles cause a thin layer of enhanced cross-polarized signal (top panel) and depolarization ratio (central panel). Daytime noise is visible in the RH panel in the first half hour after sunset at 22:06 UTC.





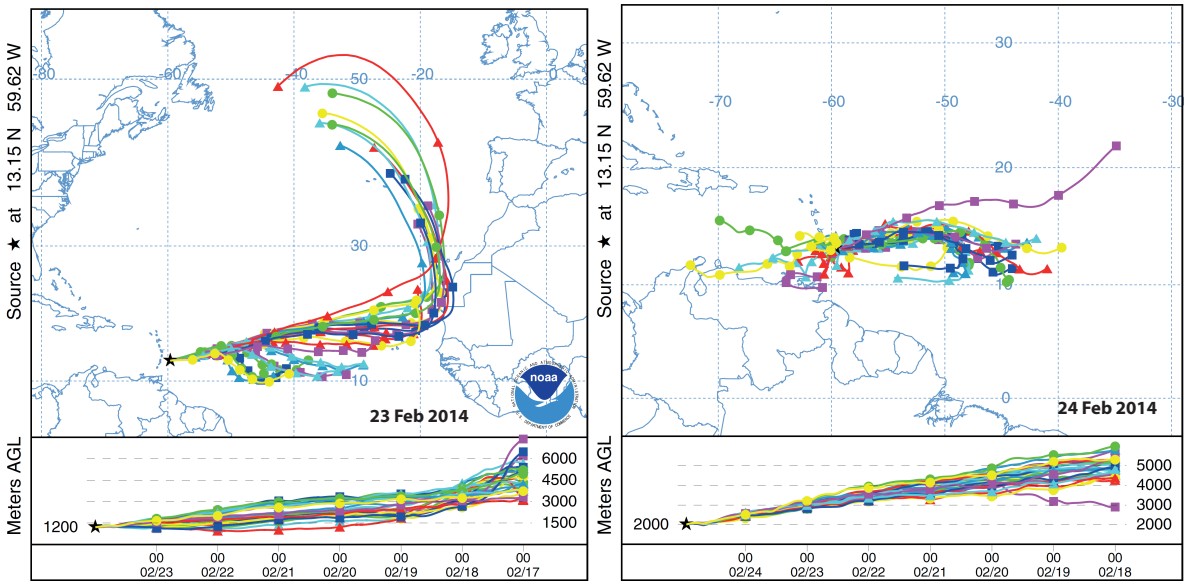

**Figure 6.** Ensemble of 7-day backward trajectories (HYSPLIT, 2017) for 24 February 2014, 00:00 UTC, arriving at 1200 m over Barbados (left) and for 24 February 2014, 23:00 UTC, arriving at 2000 m over Barbados (right).

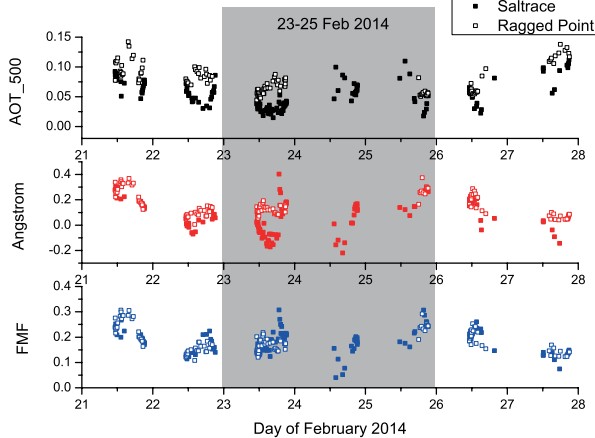

**Figure 7.** AERONET retrieval products (aerosol optical thickness at 500 nm, Ångström exponent (440–870 nm) and fine mode fraction) for 21–28 February 2014. AERONET level 2.0 data from Ragged Point and Barbados-Saltrace are shown (AERONET, 2017). For the highlighted time period the lidar measurements are shown in Fig. 3.



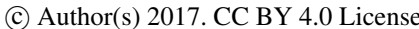


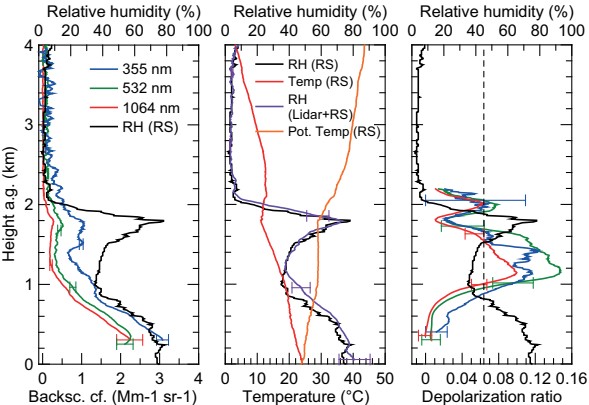

**Figure 8.** 30-minute mean profiles of particle backscatter coefficient (left) and particle linear depolarization ratio (right) at three wavelengths together with radiosonde RH (indicating the MAL up to 2 km height). The lidar observation was performed on 23 February 2014, 19:45-20:15 local time (23:45-00:15 UTC). The central panel shows the potential temperature ($T_{pot}$, radiosonde, launch at 00:00 UTC) and RH from radiosonde and from lidar (30-minute average, water vapor mixing ratio from lidar, saturation mixing ratio from radiosonde (RS)). Note the sharp drop in RH from >70% (at 1850 m height) to <10% (at 2100 m height). The dashed line in the right panel marks the sea salt efflorescence point (45% RH). Error bars in the left and right panels indicate the lidar retrieval uncertainty. The vertical smoothing window length is 50 m for the backscatter and relative humidity (lidar) and 100 m for the depolarization ratio.

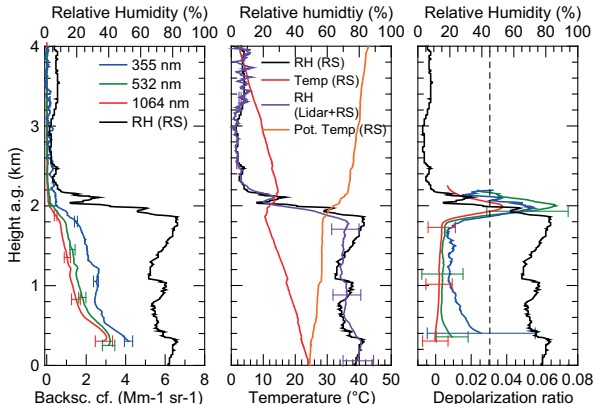

**Figure 9.** Same as Figure 8, except for 24 February 2014, 18:12-20:20 local time (22:12-00:20 UTC). In the central panel, the respective 2-hour RH profile (from lidar) is shown together with the radiosonde profiles (launch at 23:07 UTC). The MAL was entirely humid on this day.





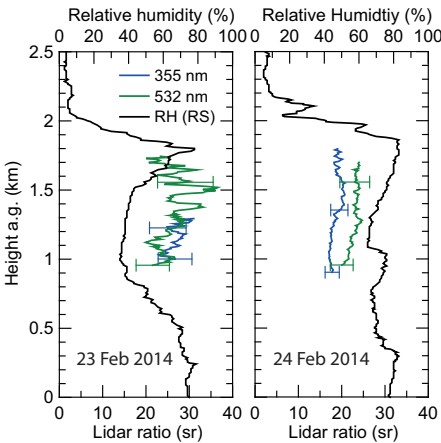

**Figure 10.** 2h mean profiles of the lidar ratio for the 23 February (left, 500 m vertical smoothing) and the 24 February 2014 (right, 750 m vertical smoothing). 532 (green line) without overlap correction, full overlap at 800–1000 m; 355 (blue line) with overlap correction, full overlap would be at 2500–3000 m. The RH profile of the radiosonde (black line) indicates the different layers.

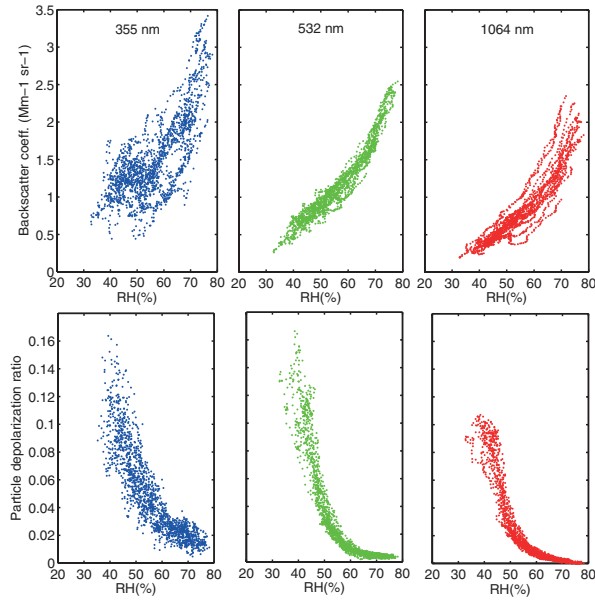

**Figure 11.** Correlation of the particle backscatter coefficient (top) and particle linear depolarization ratio (bottom) with the relative humidity for the three wavelengths 355 nm, 532 nm, and 1064 nm. The BERTHA measurement of 23 February 2014, 23:38-01:08 UTC, 375 m - 1100 m height, are used. (5 min and 50 m resolution)



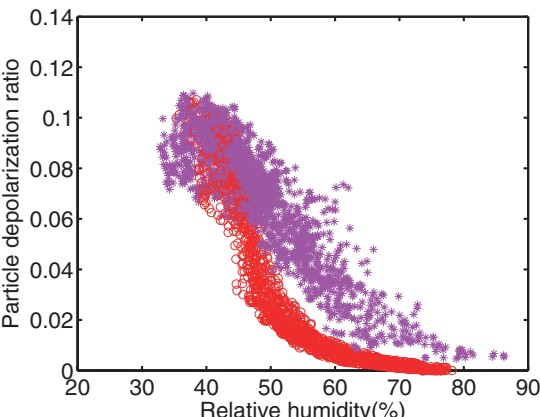

**Figure 12.** Correlation of the PLDR at 1064 nm and the relative humidity for the same settings as in Fig. 11, but for for a different height interval (375–1800 m). Above 1100 m (purple stars) the relative humidity is increasing again, up to 80%, as can be seen in Fig. 8. The depolarization ratio decreases with increasing relative humidity, but keeps higher values. The hysteresis effect between crystallization (45–48% RH) and deliquescence (70–74% RH) can be seen.

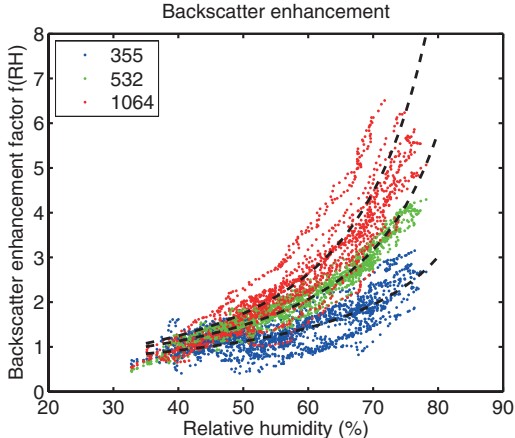

**Figure 13.** Backscatter enhancement factor for a dry value at 40% RH. The three wavelengths are fitted separately by equation (2).





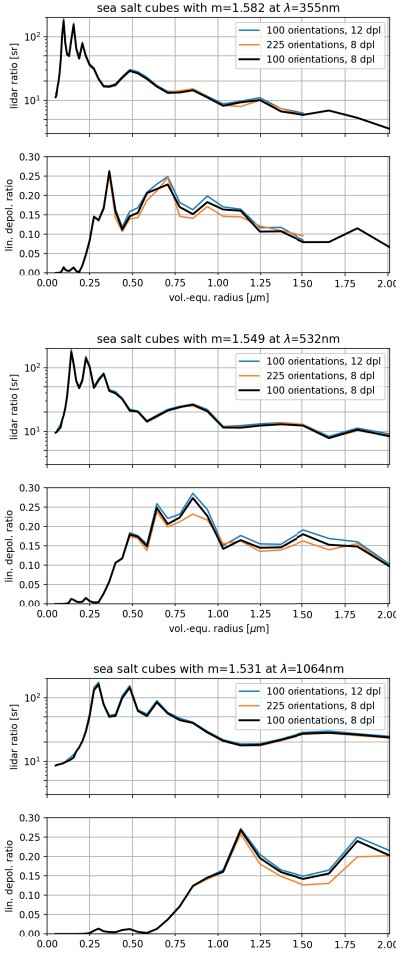

**Figure 14.** Simulation of the lidar ratio and the particle linear depolarization ratio for sea salt cubes depending on volume equivalent radius. The three lidar wavelengths are treated separately. The model simulations with increased number of dipoles (blue line) and orientations (orange line) are given as an estimation of the uncertainty. Details are given in the text.



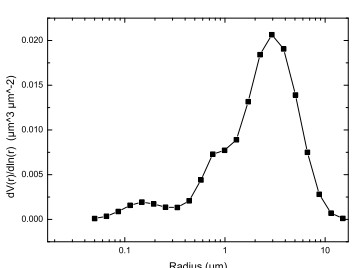

**Figure 15.** AERONET size distribution for 23 February 2014 12:31 UTC at Ragged Point, version 1.5 (AERONET, 2017).