# Peer review of "Dry versus wet marine particle optical properties: RH dependence of depolarization ratio, backscatter and extinction from multiwavelength lidar measurements during SALTRACE"

_Atmospheric Chemistry and Physics, 2017_

## Referee Comment (RC1) · Anonymous Referee #1 · 27 Jun 2017

This is an important paper that utilizes the three-wavelength polarization Raman lidar BERTHA, measuring the PLDR together with the relative humidly, to show an interesting phase shift of sea salt aerosols, form spherical to cubic like (under changing RH conditions). If indeed the measurement were done under clear marine conditions, this would have important implication for aerosol classification in remote sensing application. The authors did a careful work, with the measurements and the optical modeling as well as with the comparison and the use of other available instrumentation (radiosonde and AERONET). However, the writing needs to be

improved. It is recommended that the manuscript be accepted for publication after some minor revisions (detailed below)

General comments:

*The authors state in P9 line 13-14: "Overall we are very confident, that only pure marine aerosol was present in our measurements and no other aerosol type interfered with our measurements". The authors also show in figure 6 left panel a small cross over wet Sahara, as well as showing in figure 2 Images of dry atmospheric sea salt particles surrounded by **Saharan dust particles**. (Mentioning that these samples were taken in the dust layer 2–4 km). The reviewer agrees with the authors that it is likely that most of the aerosols are indeed sea salt. Especially if the authors claim to have ensemble trajectories below 2000m which passed only over the ocean. Nevertheless, what evidence do the authors provide for claiming there is no dust entrainment to the marine aerosol layer? In other words, how do the authors completely rule out the existence of Saharan dust, effecting their measurements? Groß et al., 2013, table 3 (1) provides the liner depolarization ratio and the LIDAR ratio, calculated for selected mixing ratios of marine aerosol and Sharan dust at 532nm. These values could be in agreement with the measurement presented in this manuscript, for example Groß et al., 2013 shows, that for 20% Saharan Dust (SD) contribution the LIDAR ratio is 21±5 (and 24±6 for 40% SD)and the liner depolarization ratio is 5±2 ( and 8±3 for 40% SD). ). The authors also present very low Ångström Exponent in this work, and even mention in P10 line 17" The drying process may occur within the marine aerosol layer or on top of it, where dry air from the free troposphere is mixed in". Is clear separation from a potential dust influence possible in this case? If not, could you add several lines in the manuscript discussing this option?

* P8 line 17-18: "...Then a fast decrease of the relative humidity... was found at the

trade wind inversion between 1850 and 2150 m. This feature was observed for most of the measurements under clean marine conditions in February 2014".

Do the authors observe the same depolarization ratio in these cases? Why only the 23 and 24 Feb are presented in this manuscript?

\* Section 5 : Could you please provide a short explanation to why the modeled lidar ratios are smaller than the measured ones. Also, could you provide a justification for the assumption that dry marine particles are halved compared to the Aeronet (assumed wet) measured size. Based on your measured backscatter values.

\*Please go over the manuscript and improve the writing. There are many small grammar/language mistakes, many of the sentences are missing punctuation marks (mainly commas), hence there are difficult to follow. Some examples: (the incorrect form is marked in Bold)

\*P2 line 10-11 : "Aerosol classification from active remote sensing (Burton et al., 2012; Groß et al., 2013) based on the depolarization ratio **will get trouble** if dried marine aerosol with a high depolarization ratio is present"

\*P2 line 28-29: "Combined with regularly available temperature profiles from radiosondes or models **we even have RH** together with the depolarization"

\*P3 line 4: "The relative humidity ranged from 40% **till** more than 80%"

\*P3 line 19:... "in a wide size range from **some** nanometer".

Example for a sentence missing punctuation marks:

\*P3 line10-11: "At the beginning we will give an introduction to sea salt aerosol under dry and humid conditions and show examples of sea salt particles collected above Barbados".

Specific minor Comments:

P4 line 10: Please add a sentence about the methodology used for this result.

Figure 2: It hard to see what the green arrows are pointing at

P6 Line 12: Please add the wavelengths

P10 Line 19-20: "There the increase in depolarization is less pronounced as in the case, where the dried marine aerosol was 20 found within the MAL (24 Feb 2014)" could you please clarify this sentence.

Technical correction:

P13 Line 14: Please correct Tab 1 to Table 1

Figure 12: Please add figure legend

(1) Groß, S., Esselborn, M., Weinzierl, B., Wirth, M., Fix, A., and Petzold, A.: Aerosol classification by airborne high spectral resolution lidar observations, Atmospheric Chemistry and Physics, 13, 2487–2505, doi:10.5194/acp-13-2487-2013, 2013.

---

## Referee Comment (RC3) · Anonymous Referee #3 · 17 Jul 2017

The paper "Dry versus wet marine particle optical properties: RH dependence of depolarization ratio, backscatter and extinction from multiwavelength lidar measurements during SALTRACE" presents a very interesting study on the effects of RH on salt particles using state-of-the-art methodologies and instrumentation. The paper is well structured and clearly written. My recommendation is publication after minor revision.

See some more detailed comments below:

P4, l16: "has been previously studied"

[Figure]

P4, l23: What is the reason to consider organics here?

P6, l24: Why is the overlap not corrected at 532 nm?

P6, l30: Please, provide estimated values for the uncertainties.

P6, l8: MAL has not been defined before.

P9, l13: Consider softening this statement. From the backward trajectories analysis and the data presented here you cannot completely assure there is no mineral dust influence on 23 February 2014. P9, l32: Why are you averaging 30 min on 23 Feb and 2 hours on 24 February?

P10, l32: Rephrase this sentence "There the increase in depolarization is less pronounced as in the case, where the dried marine aerosol was found within the MAL (24 Feb 2014)."

P11, l13: Did you check somehow that the number concentration stays the same in the two cases presented here? How? The authors should include that information on the manuscript.

P12, l32: How did you include the errors of the lidar ratio in the error of $f\alpha$?

P12, l17-25: You should include more discussion on the uncertainties here. A relative uncertainty of 12% in the relative humidity retrievals can lead to very large uncertainties in the enhancement factor (see e.g. Titos et al., 2016, and references therein). These large uncertainties can partly explain the differences with the different values provided in the literature.

G. Titos, A. Cazorla, P. Zieger, E. Andrews, H. Lyamani, M.J. Granados-Muñoz, F.J. Olmo, L. Alados-Arboledas, Effect of hygroscopic growth on the aerosol light-scattering coefficient: A review of measurements, techniques and error sources, Atmospheric Environment, 141, 494-507, ISSN 1352-2310, http://dx.doi.org/10.1016/j.atmosenv.2016.07.021, 2016.

---

## Short Comment (SC1) · 10 Aug 2017

I have read with great interest your discussion paper on the hygroscopic effects of aerosol optical properties of marine aerosol particles. A few remarks came up during the reading of the manuscript which might be helpful.

[Figure]

Classification of 'dry particles' and the effect of relative humidity on the scattering enhancement factor

The question of when a particle can be considered 'dry' is a tricky one. Thresholds of relative humidity (RH)<50 % are mentioned within the manuscript. This value is not sufficiently low to determine that a particle is dry for a number of reasons. Firstly, as has been shown in numerous studies, sea spray aerosol particles take up water at relatively low RH's (see e.g. Fig. 1 in Tang et al. (1997) or Fig. 2 in Zieger et al. (2017)). Even at RH's close to 0 % the inorganic sea salt particles will still contain water, due to the presence of hydrates which will influence their overall hygroscopic growth (Zieger et al., 2017).

In addition, the history of the RH the particles have experienced is important. If particles have experienced high values of RH, that is an RH above their main deliquescence RH, which is most likely the case if they were freshly formed in the marine boundary layer, they will stay on the upper branch of the hysteresis curve down to roughly RH≈40 %.

To better demonstrate this effect, I have plotted in Fig. 1 the calculated scattering enhancement factor $f$(RH) of pure inorganic sea salt based on measurements conducted at Stockholm University using a sea spray simulation chamber (Salter et al., 2014). The calculations were performed using Mie theory based on measured size distributions (Salter et al., 2015) and the recently determined hygroscopic growth factors (Zieger et al., 2017) as input. At RH≈50 %, the remaining water can contribute up to a factor of 2 to the scattering compared to dry conditions. But even if the particles have experienced very low RH, the contribution of water can be up to 20-30 % of the overall particle light scattering coefficient. Therefore, careful consideration of the temporal evolution of the RH that particles have experienced is critical. Again, this highlights the point that a threshold of RH=50 % is too high to classify particles as 'dry'.

The effect of particle shape

Inorganic sea salt is a complex mixture of inorganic salts and includes hydrates. This complex composition, along with the rate at which the particles have dried (see Wang et al., 2010), has implications for the shape of the 'dry' sea salt particles. As we have recently shown, inorganic sea salt particles tend to be more spherical than pure NaCl especially with increasing particle diameter (see Fig. 1 in Zieger et al., 2017). Indeed, even pure NaCl particles are not always perfect cubes (Zelenyuk et al., 2006; Zieger et al., 2017). These points should also be borne in mind.

A few more specific comments are listed below:

- Page 1, Line 19: Maybe this is a wrong reference, Tang et al. (1997) did not look specifically at the shape of sea salt particles.

- For the DDA modelling (Sect. 3.3), it would be helpful to know the assumed size parameters (mode diameters, width, etc.).

- Page 8, Line 28: I would not use the term 'water shell' for a marine aerosol particles since the majority of the chemical components (i.e. inorganic salts) will be dissolved at elevated RH.

- Page 8, Line 30: You will never be 100 % sure that only marine aerosol was present. I would add the word 'mainly' before 'marine aerosol'.

- Page 9, Line 2: Similar to the comment above, you can never fully exclude other aerosol sources although they are very unlikely. I would soften the language here.

- First paragraph on page 10: The discussion on the RH-dependency of the LR could be expanded by relating the presented observations to previous literature (e.g. Ackermann, 1998; Zieger et al., 2011).
- Page 10, line 27: Here, I would make it more clear that you have observed a specific case of the dehydration of particles along the upper branch of the hysteresis curve (i.e. that they have fully deliquesced before they are dehydrated again). In addition, one should keep in mind that inorganic sea salt has multiple eflorescence points due to its complex composition (see e.g. Tang et al., 1997; Zieger et al., 2017).

- Equation 2 and Fig. 13: The upper and lower branches of the hysteresis curve would have to be fitted separately. Therefore, one would not expect a value of $A = 1$ but rather a value of $A > 1$ for sea salt aerosol, especially if you consider that most of the particles will have fully deliquesced at one point and be on the upper branch of the hysteresis curve. Since the values of $A$ are here clearly below 1, I assume that the reference RH of the backscattering coefficient is not sufficiently low (or just not known).

- Equation 2 and Fig. 13: Here, you should also discuss that these results are only valid if the reference RH of $40 - 50\%$ is sufficient to represent 'dry' particles, which, as discussed above, is probably not the case. This will be a critical point for modellers who will relate their 'wet' values of aerosol optical properties to perfectly dry values (at RH=0%). As such, they will subsequently estimate much higher enhancement factors than shown in Table 2 or Fig. 13. Figure 1 below shows the same $\gamma$-fit as used in the presented study. Extrapolating to RH=0 % will give a value of approx. $A \approx 1.5$ (with $\gamma \approx 0.45$).

**References**

Ackermann J., The extinction-to-backscatter ratio of tropospheric aerosol: A numerical study, *J. Atmos. Oceanic Technol.*, 15(4), 1043–1050, 1998.
Salter M.E., Nilsson E.D., Butcher A., and Bilde M., On the seawater temperature dependence

of the sea spray aerosol generated by a continuous plunging jet, *J. Geophys. Res.*, 119(14), 9052–9072, doi:10.1002/2013JD021376, 2014.

Salter M.E., Zieger P., Acosta Navarro J.C., Grythe H., Kirkevåg A., Rosati B., Riipinen I., and Nilsson E.D., An empirically derived inorganic sea spray source function incorporating sea surface temperature, *Atmos. Chem. Phys.*, 15(19), 11047–11066, doi:10.5194/acp-15-11047-2015, 2015.

Tang I.N., Tridico A., and Fung K., Thermodynamic and optical properties of sea salt aerosols, *J. Geophys. Res.*, 102(D19), 23269–23275, 1997.

Wang Z., King S.M., Freney E., Rosenoern T., Smith M.L., Chen Q., Kuwata M., Lewis E.R., Pöschl U., Wang W., Buseck P.R., and Martin S.T., The Dynamic Shape Factor of Sodium Chloride Nanoparticles as Regulated by Drying Rate, *Aerosol Sci. Tech.*, 44, 939–953, doi:10.1080/02786826.2010.503204, 2010.

Zelenyuk A., Cai Y., and Imre D., From agglomerates of spheres to irregularly shaped particles: Determination of dynamic shape factors from measurements of mobility and vacuum aerodynamic diameters, *Aerosol Sci. Technol.*, 40(3), 197–217, 2006.

Zieger P., Väisänen O., Corbin J., Partridge D.G., Bastelberger S., Mousavi-Fard M., Rosati B., Gysel M., Krieger U., Leck C., Nenes A., Riipinen I., Virtanen A., and Salter M., Revising the hygroscopicity of inorganic sea salt particles, *Nature Communications*, 8(15883), doi:10.1038/ncomms15883, 2017.

Zieger P., Weingartner E., Henzing J., Moerman M., de Leeuw G., Mikkilä J., Ehn M., Petäjä T., Clémer K., van Roozendael M., Yilmaz S., Frieß U., Irie H., Wagner T., Shaiganfar R., Beirle S., Apituley A., Wilson K., and Baltensperger U., Comparison of ambient aerosol extinction coefficients obtained from in-situ, MAX-DOAS and LIDAR measurements at Cabauw, *Atmos. Chem. Phys.*, 11(6), 2603–2624, doi:10.5194/acp-11-2603-2011, 2011.

**Fig. 1.** Scattering enhancement of inorganic sea salt calculated using Mie theory (see text for details).

---

## Author Comment (AC1) · 23 Sep 2017

**Letter of Reply to Referee 1**

Thank you for carefully reading the manuscript and providing useful suggestions to improve the paper.
The changes in the manuscript are marked in bold.
To strengthen the point of pure marine conditions during our measurements, we added a more sophisticated air mass analysis (Fig. 7), which takes different land covers into account. Martin Radenz who performed the calculations was added as a co-author.

*General comments:*

*\*The authors state in P9 line 13-14: "Overall we are very confident, that only pure marine aerosol was present in our measurements and no other aerosol type interfered with our measurements". The authors also show in figure 6 left panel a small cross over west Sahara, as well as showing in figure 2 Images of dry atmospheric sea salt particles surrounded by Saharan dust particles. (Mentioning that these samples were taken in the dust layer 2–4 km). The reviewer agrees with the authors that it is likely that most of the aerosols are indeed sea salt. Especially if the authors claim to have ensemble trajectories below 2000m which passed only over the ocean. Nevertheless, what evidence do the authors provide for claiming there is no dust entrainment to the marine aerosol layer? In other words, how do the authors completely rule out the existence of Saharan dust, effecting their measurements? Groß et al., 2013, table 3 (1) provides the liner depolarization ratio and the LIDAR ratio, calculated for selected mixing ratios of marine aerosol and Sharan dust at 532nm. These values could be in agreement with the measurement presented in this manuscript, for example Groß et al., 2013 shows, that for 20% Saharan Dust (SD) contribution the LIDAR ratio is 21±5 (and 24±6 for 40% SD) and the linear depolarization ratio is 5±2 ( and 8±3 for 40% SD). ). The authors also present very low Ångström Exponent in this work, and even mention in P10 line 17" The drying process may occur within the marine aerosol layer or on top of it, where dry air from the free troposphere is mixed in". Is clear separation from a potential dust influence possible in this case? If not, could you add several lines in the manuscript discussing this option?*

**It is good, that you point out this issue. So, we strengthened our discussion to support the existence of pure marine conditions. Figure 7 and some paragraphs have been added (p4, l20-23; p8, l10-12; p9, l11-32).**

**The dry air from the free troposphere mixed in at the MAL top must be clean, because the total backscatter signal is weaker in the upper 200 m. Therefore, the dry air dilutes the marine aerosol layer. If there was 40% Saharan dust contribution, the signal would be much stronger. And there is no indication of the presence of Saharan dust above the trade wind inversion height in our measurements, as it is the case during the summer months, where a Saharan air layer can be observed (Groß et al., ACP 2015, Haarig et al., 2017). Dust transport over 5000 km within the MAL is very unlikely as dust is removed efficiently from the MAL (Rittmeister et al., ACP 2017).**

**Now, Fig. 7 shows the relative residence time close to the ground (<2 km height) of the ensemble of trajectories prior to their arrival over Barbados. It can be seen, that aerosol load has taken place over the ocean only.**

*\* P8 line 17-18: ". . .Then a fast decrease of the relative humidity... was found at the trade wind inversion between 1850 and 2150 m. This feature was observed for most of the measurements under clean marine conditions in February 2014". Do the authors observe the same depolarization ratio in these cases? Why only the 23 and 24 Feb are presented in this manuscript?*

**Enhanced depolarization ratios (between 0.04 and 0.12 at 532 nm) at the other days were observed. A sentence to state this was added (p8, l14-16). The two presented cases are the most pronounced. The 24 February clearly shows the enhanced depolarization ratio at MAL top and the 23 February offers the opportunity to study the drying process over a larger height range. Therefore, these two cases have been used as case studies.**

*\* Section 5 : Could you please provide a short explanation to why the modeled lidar ratios are smaller than the measured ones.*

**From the modeling side, it is clear that the low lidar ratio of the bulk aerosol comes from the very low lidar ratios of large salt cubes (e.g., the lidar ratio is 3.6 sr for r=2 µm at 355 nm as shown in Fig. 15). There are several potential reasons for the discrepancy of the bulk aerosol:**

**a) Too many large particles in our model.**

**b) Real dry sea salt particles are not perfect cubes and the real lidar ratio of large dry sea salt particles is larger than for cubes.**

**c) The refractive index is wrong.**

**d) DDA calculations are wrong.**

**As we used size distributions integrated over the whole atmospheric column and we do not have specific information on the size distribution in the dry sea salt layer, it is not unlikely that our model size distribution deviates from the real size distribution. Removing particles larger than 1-2 µm from the bulk improves the agreement with the measured lidar ratios. Thus reason a) is not unlikely but we have no further information on size distributions.**

**From our previous work, we have the feeling that low lidar ratios are a general feature of large non-absorbing particles (at least in the size parameter range from about 10 to 100) if their aspect ratio is small, or in other words if they are compact. For example, for oblate and prolate spheroids with aspect ratio 1.2 (m=1.582) we get 3.3 sr and 5.1 sr for r=2 µm at 355 nm, which is close to the value we got for the cubes.**

**Therefore we think reason b) is unlikely because low aspect ratios are characteristic for sea salt particles.**

**Regarding the refractive index, we find that the lidar ratio is not strongly dependent on m within the range covered by our simulations (1.531-1.582) making reason c) unlikely.**

**We tried other DDA formulations included in ADDA, but also with them we get lidar ratios <5 sr for the large particles mentioned above. Thus also d) seems unlikely to us.**

*Also, could you provide a justification for the assumption that dry marine particles are halved compared to the Aeronet (assumed wet) measured size. Based on your measured backscatter values.*

**Hygroscopic growth factors of 2 (ranging between 1.77 and 2.14) are reported for sea salt (review by Swietlicki et al., 2008). This is the basis for our assumption, that dry sea salt particles have only half the radius of wet sea salt particles.**

*\*Please go over the manuscript and improve the writing. There are many small grammar/ language mistakes, many of the sentences are missing punctuation marks (mainly commas), hence there are difficult to follow. Some examples: (the incorrect form is marked in Bold)*

**The manuscript was checked again, and commas have been added.**

*P2 line 10-11 : "Aerosol classification from active remote sensing (Burton et al., 2012; Groß et al., 2013) based on the depolarization ratio **will get trouble** if dried marine aerosol with a high depolarization ratio is present"*

**"will be misleading"**

*P2 line 28-29: "Combined with regularly available temperature profiles from radiosondes or models **we even have RH** together with the depolarization"*

**"our Raman lidar observations provide RH"**

*P3 line 4: "The relative humidity ranged from 40% **till** more than 80%"*

**"from 40% to more than 80%"**

*P3 line 19:. . . "in a wide size range from **some** nanometer".*

**"in a wide size range from nanometer up to several micrometer"**

*Example for a sentence missing punctuation marks:*

*P3 line10-11: "At the beginning we will give an introduction to sea salt aerosol under dry and humid conditions and show examples of sea salt particles collected above Barbados".*

**Comma added.**

*Specific minor Comments:*

*P4 line 10: Please add a sentence about the methodology used for this result.*

**The SEM used for this study is now mentioned (in the caption of Fig. 1 as well).**

*Figure 2: It hard to see what the green arrows are pointing at*

**Now the green arrows in Fig. 2 are larger and slightly brighter and point to the outline of the evaporated droplet.**

*P6 Line 12: Please add the wavelengths*

**The three wavelengths are added.**

*P10 Line 19-20: "There the increase in depolarization is less pronounced as in the case, where the dried marine aerosol was 20 found within the MAL (24 Feb 2014)" could you please clarify this sentence.*

**The whole paragraph was deleted.**

*Technical correction:*

*P13 Line 14: Please correct Tab 1 to Table 1*

**Changed.**

*Figure 12: Please add figure legend*

**A legend is added to the figure.**

[revised manuscript text omitted]

---

## Author Comment (AC2) · 23 Sep 2017

**Letter of Reply to Referee 2**

Thank you for carefully reading the manuscript and providing useful suggestions to improve the paper. The changes in the manuscript are marked in bold.
To strengthen the point of pure marine conditions during our measurements, we added a more sophisticated air mass analysis (Fig. 7), which takes different land covers into account. Martin Radenz who performed the calculations was added as a co-author.

*Specific Comments*

*Figure 2 is very interesting and helpful for visualizing the sea salt particles at different stages of the transition. Initially the dust in these images confused me as I was concerned that it might be very hard to separate dry sea salt from dust if they are mixed together like this. On more careful reading, I see that the images in Figure 2 are from a completely different time of year to the case studies. This is perfectly fine, but you might consider putting in another sentence or two in key places to make sure that it is very clear (even to people who are reading a little too fast) that the images in Figure 2 are not from the same time period as the main case studies in the paper.*

**Thank you very much for this comment. Two phrases on p4, l19-23 and in the caption of Fig 2 have been added to make sure, that it is very clear for everyone.**

*Page 5, line 9-14 is confusing about whether spherical or non-spherical sea salt particles would be expected to be bigger. Possibly this confusion would be removed if you make a clearer distinction between humidified sea salt particles and droplets. Is it true that crystals are larger than solution droplets, but then the crystals continue to grow as they are re-humidified, so that humidified crystals are larger than solution droplets? Or am I misunderstanding the text in this paragraph?*

**To clarify this point: In general the sea salt solution droplets are larger than the sea salt crystals, because the humidified sea salt particles (=solution droplets) grow due to water uptake.**

**The mode radii from Sakai et al., 2010, have been removed from the manuscript. Their high standard deviation (0.08 ± 2.1 µm for the droplets and 0.19 ± 1.8 µm for the crystals) hampers a clear conclusion, which particles are larger in this laboratory study. They prepared the particles in five radius ranges between 0.15 and 2.5 µm (Sakai et al, 2010). Instead the standard deviations for the depolarization ratios have been added (p5, l12-14).**

*The quantities shown in the backscatter curtains Fig 3-5 are not the best ones to show. It would be more informative to show geophysical quantities, the same ones shown in Figs 8 and 9. Specifically total true backscatter coefficient rather than attenuated backscatter or the cross-polarized component of attenuated backscatter, and particulate depolarization ratio rather than volume depolarization ratio. What are the horizontal and vertical resolution of the lidar data shown in the figures?*

**Indeed, the total backscatter would convey more information, but the presented scenario (Fig. 3) will lose information due to a larger smoothing, that is necessary for the calculation of the backscatter. In Fig. 3 the range resolution of the lidar (7.5 m) is used with a time resolution of 10 s. We decided to show the high resolution plot of the range-corrected cross polarized signal at 1064 nm as it is very sensitive to aerosol particles.**

The volume linear depolarization ratio at 1064 nm is shown in Fig. 4 and 5. The molecular contribution at this large wavelength is almost negligible, so that the shown volume depolarization ratio is almost equal to the particle depolarization ratio. This is not the case at 355 or 532 nm. In Haarig et al., ACP 2017 we discussed the advantages of measuring the depolarization ratio at 1064 nm.

The vertical and temporal resolution was added in the captions of Fig 3-5.

*Figure 10 shows a larger lidar ratio for the dry particles (lower RH). Can you explain in general terms in the paper why this is, in terms of the relative size and refractive indices of the particles?*

The difference of the lidar ratio at 532 nm is not significant. The higher lidar ratio at 355 nm for dry particles may be explained by the fact, that the dry sea salt particles are smaller. Absorption is very weak for sea salt particles (high single scattering albedo). Therefore, the extinction is caused almost only by scattering. The lidar ratio is then the ratio of scattering to backscattering. Smaller particles have a larger amount of sideward scattering leading to a larger lidar ratio. A sentence has been added p10, l18-19.

*Page 11, lines 10-13 states that particle backscatter coefficient is an indicator of particle growth if the number concentration stays the same. However, there is no particular reason to imagine the number concentration is the same for these observations, is there? However, you have two wavelengths of extinction, and Angstrom exponent of extinction is a more usual proxy for particle size (indeed you use AOT Angstrom exponent from AERONET in Fig 7). It would be useful to show extinction Angstrom exponent from the lidar measurements to see if this supports the idea of particle growth.*

The extinction Angstrom exponent cannot be derived satisfactory below 1 km due to overlap issues. Still the extinction Angstrom exponent is very uncertain, as the extinction is small. Between 1.0 and 1.5 km height (the dry marine layer) the extinction Angstrom exponent (E355/E532) increases indicating a decreasing particle size. The same holds for the backscatter Angstrom exponent (B532/B1064), which can be derived at lower altitudes as well. A sentence was added p11, l12-14. The profiles of the Angstrom exponent are shown here, but not in the paper:

[Figure]

**Figure 1** Vertical profiles of the Angstrom exponents measured 23 Feb 2014 23:38-01:38 UTC with uncertainty.

*Same place. I also think that particle backscatter coefficient has a dependence on refractive index which may play a role in the trends seen here, although I don't know the relative strengths*

*of the dependencies on N, refractive index and particle size in this observation. Can you comment on this?*

**We cannot measure the refractive index. Simulations show, that the lidar ratio (extinction-to-backscatter ratio) is independent of refractive index in the observed size range.**

*Page 12 line 1-2 discusses a poor agreement when extrapolating. It's not surprising that extrapolating from an exponential fit might produce poor agreement. It would be useful to show a linearized version of Figure 13 (that is, log of enhancement factor vs log(1-RH/100) ) which may make it more obvious in what regime the parameterization of Eq. 2 holds or might even actually improve the fit and the extrapolation, depending on how the parameters were fitted (that is, was the fit done on the linearized system?)*

**We added the suggested log-log plot in Fig. 14. It can be seen from this plot that the exponential (now linear) fit does not hold for the highest and especially for the lowest RH values. Two sentences are added (p12, l6-8).**

*Page 12 line 10. What does "modeled" mean? Is this the simulated data in Table 1 that is explained much later?*

**The simulated lidar ratios (Table 1) are used. A reference to the table was added (p12, l20).**

*Page 12 line 15. I don't understand this. The increase 2.05 to 3.73 to 5.37 is more than 8% from one wavelength to the next. "Zieger et al stated that the wavelength dependence for marine aerosol". Do you mean the wavelength dependence of the enhancement factor?*

**The 8% increase from one wavelength to the next was observed by Kotchenruther et al., 1999 for 450, 550 and 700 nm. The number was deleted. In the next sentence, the wavelength dependence is referring to the scattering enhancement factor for marine aerosol. The passage was deleted.**

*Same place: it seems that you could infer the expected behavior of the wavelength dependence by assuming that the angstrom exponent depends more or less linearly on size and use standard parametrization of the hygroscopic growth of particles. Would that be a reasonable assumption and would such a calculation give you similar wavelength dependence to what you see?*

**The small (dry) particles' influence is less pronounced for the long wavelengths (1064 nm) compared to the UV. The particle grows due to water uptake and gets more relevant in the scattering process for the long wavelengths as well. This dependence can be seen in Fig. 12 a-c. The backscatter at the dry RH is much smaller for 1064 nm. Thus, the enhancement is much stronger than at 355 nm.**

*Figure 14 and section 5. It would be helpful to mark a line on each panel of Figure 14 showing the value of volume equivalent radius used for the calculation of simulated results in Table 1 and to include this information also in Table 1.*

**For the calculations we used all values shown in Fig. 15. The different radii were weighted according to the AERONET size distribution (or an additional case assuming halved size because of drying). We make this point now clearer in Section 5. Additionally, we added the effective radius (r=1.033 μm) of the AERONET size distribution to the simulations of LR and PLDR in Fig. 15.**

*Page 13, lines 16-18. I got distracted trying to understand if the authors mean this last sentence of the paragraph to be related to the rest of the paragraph or not. If it is an unrelated observation, I suggest moving it into its own paragraph. If it is related, I suggest adding some kind of transition to make the reason for making this statement more obvious. Perhaps just*

*"However" or "Nevertheless", that is, if you are meaning to suggest that we need not worry overmuch about the disagreement between measured and modeled lidar ratios since it nevertheless represents a relatively small variation in the properties of the type compared to differences between types.*

**We followed your suggestion and added "Nevertheless" on p14, l2 and reshaped the paragraph.**

*Page 14, There are some rather vague and unconvincing conclusions here. It seems like you are struggling a little to define the relevance of the observation. I first want to say that this is a very interesting and publishable observation and analysis even in the worst case if it is not particularly relevant to other specific calculations like radiative transfer or aerosol types. If these arguments can't be strengthened, it would be better to take them out with no loss to the paper, in my opinion. More specifically, first, it's not clear to me how or whether this observation affects radiative transfer calculations. What is the mechanism by which dried sea salt aerosol would have a different radiative effect than humidified sea salt in a way that is not captured in the measurements? The differences in backscatter or extinction are directly measured so whether the backscattering comes from depolarizing particles or not is not relevant. Is it because of a poor inference of the refractive index? But you hardly discuss refractive index in this paper. Likewise, what is the mechanism (what characteristic of the dried sea salt) that would cause an error in the AERONET retrievals? Moreover, even if these layers are very common (which it is not at all obvious to me), the layer you observed is very thin compared to the aerosol below it. How much effect would this realistically have on a full column calculation of radiative transfer? The authors could do a rough, approximate calculation to show whether this is a noticeable effect or not, and then I might be more convinced. It also would be good to expand the discussion of aerosol typing to explain why it would matter to analysis such as the cited references if a small amount of aerosol is mistyped. Is this specifically about inferring lidar ratios for aerosol extinction retrievals, like CALIPSO, or is there some other implication to mistyping?*

**Thank you for your detailed discussion about the conclusion. We reshaped and shortened the conclusion.**

**The layer of dried marine aerosol is thin compared to the thicker layer of humid marine aerosol, so the radiative effect might not be that strong (p14, l22-24). But it would be interesting to see the effect in the models. The discussion is not about the radiative effect in the measurements but in the radiative transfer models, which calculate for example the radiative forcing by aerosol layers. We agree that the effect on backscatter and extinction is captured with our lidar measurements.**

**Inversion algorithms have to assume a shape for the particles. Due to simplicity, a spherical shape is usually assumed. It has been shown that this does not work for non-spherical dust particles. To better invert dusty scenarios, a spheroidal shape is assumed. It would not be difficult to assume a cubic shape as well to mimic non-spherical dry marine aerosol. We softened the text concerning AERONET and suggest using a cubic model (p14, l25-27). It would be interesting to see for some test cases, how a mixture of cubes and spheres would perform.**

**The problems in mistyping aerosol types are further discussed (p14, l28 – p15, l2). It is not only to derive extinction values for satellites. Recently estimates of mass, CCN and INP concentration profiles are provided from lidar measurements (Mamouri and Ansmann, 2016). The depolarization ratio plays a key role in these retrievals and dry marine particles with a high depolarization ratio will cause errors in the retrievals.**

**Furthermore, we expressed the wish that global studies with satellites (CALIPSO or EarthCARE) would estimate the global occurrence of dry marine aerosol layer (p14, l21-22). In our own lidar measurements, we found further evidence over the Southern**

**Atlantic, over Cyprus and Haifa. Measurements over the Ocean are rare, so it is not that easy to address the phenomena.**

*Technical comments*

*The abstract seems a little scattered. Each sentence seems to stand alone without clear relationships between them. Rearranging the sentences to be in a more logical order and adding some transitional phrases ("therefore", "however", etc.) or combining sentences would help.*

**The abstract was rearranged and shorted to be more convenient.**

*Page 6, line 5. "See AERONET web page, Barbados-SALTRACE site". Please give specific URLs.*

**The URL for the AERONET web page is given. The station name is Barbados_SALTRACE.**

*What wavelength is Fig 3-5? Please include the information in the figure labels or captions.*

**The wavelength information (1064 nm) is indicated in each figure and in the caption.**

*Figure 8. From the text, I think this is particulate depolarization ratio (rather than volume). Please confirm, and include that information in the caption.*

**The caption of the axis has been changed to explicitly state "Part. depolarization ratio".**

*Figures 8 and 9. It seems there is meant to be a dashed vertical line in all 3 panels, but it appears only in the right panels.*

**The crystallization RH line has been added in all 3 panels. Additionally, we added dashed lines at the crystallization RH in Fig. 11-13.**

*Bigger axis labels and legends are needed in the figures, big enough to be on the order of other font sizes used in the paper*

**Thank you for this hint. The font sizes in the figures have been increased.**

[revised manuscript text omitted]
=80%) | **19 ± 5** | 23 ± 2 | – | ≤ 3.0 | ≤ 2.0 | ≤ 2.0 | 2014-02-24 22:11–00:21 UTC, <1800 m |
| **Dry** (RH=40%) | **27 ± 6** | 25 ± 3 | – | **11.5 ± 8.2** | **14.8 ± 3.5** | **9.9 ± 1.1** | 2014-02-23 23:38–01:08 UTC, **1150 m** |
| **Simulation** | | | | | | | |
| **Wet** (Spherical) | 22 ± 2 | 27 ± 3 | 35 ± 4 | 0 | 0 | 0 | **Mie**, Aeronet SD 2014-02-23 |
| **Dry** (Cubic) | 10 ± 1 | 16 ± 2 | **31 ± 3** | **8.8 ± 2.0** | **11.9 ± 2.0** | **14.2 ± 2.0** | DDA, Aeronet SD 2014-02-23 |
| | 13 ± 1 | **20 ± 2** | 36 ± 4 | **10.8 ± 2.0** | **12.8 ± 2.0** | **12.0 ± 2.0** | DDA, Aeronet SD 2014-02-23, **radius/2** |

[revised manuscript text omitted]

---

## Author Comment (AC3) · 23 Sep 2017

**Letter of Reply to Referee 3**

Thank you for carefully reading the manuscript and providing useful suggestions to improve the paper. The changes in the manuscript are marked in bold.
To strengthen the point of pure marine conditions during our measurements, we added a more sophisticated air mass analysis (Fig. 7), which takes different land covers into account. Martin Radenz who performed the calculations was added as a co-author.

*See some more detailed comments below:*

*P4, l16: "has been previously studied"*

**The adverb is moved to the correct position.**

*P4, l23: What is the reason to consider organics here?*

**They are considered, because organics will lead to a less crystalline and therefore more spherical shape. Sea salt emitted from the ocean might contain organics of marine origin. A reference and "the marine origin" have been added (p4, l27-28).**

*P6, l24: Why is the overlap not corrected at 532 nm?*

**The 532 nm channel reaches full overlap at 800-1000 m height (information added on p6, l24-26), whereas the 355 nm channel reaches in this configuration full overlap between 2500 and 3000 m. For the observation of the dry sea salt particles at 1100-1300 m no overlap correction is necessary at 532 nm.**

*P6, l30: Please, provide estimated values for the uncertainties.*

**The estimation of the uncertainties is described in detail by Haarig et al., 2017. For the volume depolarization ratio the uncertainty at 355 nm is 0.01 (information added on p7, l1). Considering a large molecular contribution in the UV, this leads to uncertainties in the particle linear depolarization ratio of up to 0.08.**

*P6, l8: MAL has not been defined before.*

**The passage is rephrased to avoid the abbreviation "MAL".**

*P9, l13: Consider softening this statement. From the backward trajectories analysis and the data presented here you cannot completely assure there is no mineral dust influence on 23 February 2014.*

**A passage on the usual way of dust transport was added (p9, l25-29) and the statement was softened (now at p9, l29-31). Nevertheless, we are very confident that we had pristine marine conditions. The new Figure 7 supports the predominance of marine aerosols.**

*P9, l32: Why are you averaging 30 min on 23 Feb and 2 hours on 24 February?*

**The scenario is changing much faster at 23 February 2014 as shown in Fig. 3. At 24 February 2014, the scenario stayed the same over the 2 hours of measurement as shown in Fig. 4. In order to not average over changing atmospheric conditions we have chosen different averaging times. A sentence to explain the 30 min average was added in the caption of Fig. 9.**

*P10, l32: Rephrase this sentence "There the increase in depolarization is less pronounced as in the case, where the dried marine aerosol was found within the MAL (24 Feb 2014)."*

**The whole paragraph was deleted.**

*P11, I13: Did you check somehow that the number concentration stays the same in the two cases presented here? How? The authors should include that information on the manuscript.*

**We cannot check it, but we see in the increase of the extinction and backscatter Angstrom exponent that the particle size decreases with decreasing RH. This is a better indication for particle growth. The information has been added on p11, l12-14.**

*P12, I32: How did you include the errors of the lidar ratio in the error of f_alpha?*

**The relative errors of the lidar ratios and the relative errors of f_beta are included by Gaussian error propagation. A note was included on p12, l19.**

*P12, I17-25: You should include more discussion on the uncertainties here. A relative uncertainty of 12% in the relative humidity retrievals can lead to very large uncertainties in the enhancement factor (see e.g. Titos et al., 2016, and references therein). These large uncertainties can partly explain the differences with the different values provided in the literature.*

**Thank you for pointing to the Titos et al., 2016 publication. The error of the relative humidity is now included in the errors of f_beta and f_alpha by looking at the range of RH 40±5% and 80±10%. Overall, we get a relative error in the scattering enhancement factor of 50% in the UV and 30% at 532 and 1064 nm. The manuscript was changed at the corresponding passages (p12, l13; p12, l21-22 and Table 2).**

[revised manuscript text omitted]
=80%) | **19 ± 5** | 23 ± 2 | – | ≤ 3.0 | ≤ 2.0 | ≤ 2.0 | 2014-02-24 22:11–00:21 UTC, <1800 m |
| **Dry** (RH=40%) | **27 ± 6** | 25 ± 3 | – | **11.5 ± 8.2** | **14.8 ± 3.5** | **9.9 ± 1.1** | 2014-02-23 23:38–01:08 UTC, **1150 m** |
| **Simulation** | | | | | | | |
| **Wet** (Spherical) | 22 ± 2 | 27 ± 3 | 35 ± 4 | 0 | 0 | 0 | **Mie**, Aeronet SD 2014-02-23 |
| **Dry** (Cubic) | 10 ± 1 | 16 ± 2 | **31 ± 3** | **8.8 ± 2.0** | **11.9 ± 2.0** | **14.2 ± 2.0** | DDA, Aeronet SD 2014-02-23 |
| | 13 ± 1 | **20 ± 2** | 36 ± 4 | **10.8 ± 2.0** | **12.8 ± 2.0** | **12.0 ± 2.0** | DDA, Aeronet SD 2014-02-23, **radius/2** |

[revised manuscript text omitted]

---

## Author Comment (AC4) · 23 Sep 2017

**Letter of Reply to Paul Zieger**

Thank you for carefully reading the manuscript and providing useful suggestions. It helped us to improve the paper, especially the discussion about 'dry' sea salt particles. The changes in the manuscript are marked in bold.

To strengthen the point of pure marine conditions during our measurements, we added a more sophisticated air mass analysis (Fig. 7), which takes different land covers into account. Martin Radenz who performed the calculations was added as a co-author.

*Classification of 'dry particles' and the effect of relative humidity on the scattering enhancement factor*

*The question of when a particle can be considered 'dry' is a tricky one. Thresholds of relative humidity (RH)<50% are mentioned within the manuscript. This value is not sufficiently low to determine that a particle is dry for a number of reasons. Firstly, as has been shown in numerous studies, sea spray aerosol particles take up water at relatively low RH's (see e.g. Fig. 1 in Tang et al. (1997) or Fig. 2 in Zieger et al. (2017)).*

*Even at RH's close to 0% the inorganic sea salt particles will still contain water, due to the presence of hydrates which will influence their overall hygroscopic growth (Zieger et al., 2017).*

*In addition, the history of the RH the particles have experienced is important. If particles have experienced high values of RH, that is an RH above their main deliquescence RH, which is most likely the case if they were freshly formed in the marine boundary layer, they will stay on the upper branch of the hysteresis curve down to roughly RH_40 %.*

*To better demonstrate this effect, I have plotted in Fig. 1 the calculated scattering enhancement factor f(RH) of pure inorganic sea salt based on measurements conducted at Stockholm University using a sea spray simulation chamber (Salter et al., 2014). The calculations were performed using Mie theory based on measured size distributions (Salter et al., 2015) and the recently determined hygroscopic growth factors (Zieger et al., 2017) as input. At RH_50 %, the remaining water can contribute up to a factor of 2 to the scattering compared to dry conditions. But even if the particles have experienced very low RH, the contribution of water can be up to 20-30% of the overall particle light scattering coefficient. Therefore, careful consideration of the temporal evolution of the RH that particles have experienced is critical. Again, this highlights the point that a threshold of RH=50% is too high to classify particles as 'dry'.*

Thank you for your contribution to the discussion. We discuss the issue of 'dry' sea salt particles in Section 4.3 (p11, l20-29), but we will keep the term 'dry' when we refer to 40% RH, as it is used to normalize the data in different studies. It is below the crystallization point as we see by the transition in particle shape observed with the depolarization ratio. In our measurement under atmospheric conditions we cannot control RH, so we use the lowest measured RH.

It would be possible to calculate from Eq. 1 and 2 the backscatter at 0% RH and to normalize the data by this value. This would imply the assumption, that the observed shrinking will continue down to 0% RH, which is not very likely. So we have to live with the dry value at 40% RH, but we can point out the problems with this definition.

For the studies of the scattering enhancement factor, we use the value of 40% RH as dry and mention p11, l28-29 that the sea salt particles are not completely dry. 50% RH is not used as a threshold, but it can be seen in Fig. 12 that at RH=50% the depolarization ratio starts to increase significantly. If the ambient RH drops below 50% in marine

environments, there will be the possibility to have crystalline sea salt particles with different optical properties. Actually, even below 70% RH cubic-like sea salt might occur due to hysteresis effects. The message is directed to aerosol classification schemes and lidar retrievals using the depolarization ratio as a key parameter to identify and separate different aerosol types. Till now, they have used thresholds like 0.05 for the particle depolarization ratio to identify marine aerosol. This holds for marine environments with RH>70% and in many cases even for RH>50% (upper branch of the hysteresis). Below 50% they should definitely take care with their classification. The optical properties are changing even if not all particles in an air parcel are completely dry. It would be interesting to observe further drying in the atmosphere and see how the parameters especially the depolarization ratio changes (p11, l1-4).

*The effect of particle shape*

*Inorganic sea salt is a complex mixture of inorganic salts and includes hydrates. This complex composition, along with the rate at which the particles have dried (see Wang et al., 2010), has implications for the shape of the 'dry' sea salt particles. As we have recently shown, inorganic sea salt particles tend to be more spherical than pure NaCl especially with increasing particle diameter (see Fig. 1 in Zieger et al., 2017). Indeed, even pure NaCl particles are not always perfect cubes (Zelenyuk et al., 2006; Zieger et al., 2017). These points should also be borne in mind.*

Thank you. The shape discussion in Section 2 has been extended and your suggested references have been added (p4, l5-9).

*A few more specific comments are listed below:*

*• Page 1, Line 19: Maybe this is a wrong reference, Tang et al. (1997) did not look specifically at the shape of sea salt particles.*

The reference was changed to Wise et al., 2007, who observed the change in shape of NaCl particles.

*• For the DDA modelling (Sect. 3.3), it would be helpful to know the assumed size parameters (mode diameters, width, etc.).*

We did not use a parameterized size distribution (like log-normal) but used a binned size distribution provided by AERONET. This information has been added in the model description. We added the effective radius of the AERONET size distribution to the legend of Tab.1.

*• Page 8, Line 28: I would not use the term 'water shell' for a marine aerosol particle since the majority of the chemical components (i.e. inorganic salts) will be dissolved at elevated RH.*

Thank you. We agree and omit the term "water shell" on p9, l4.

*• Page 8, Line 30: You will never be 100% sure that only marine aerosol was present. I would add the word 'mainly' before 'marine aerosol'.*

We changed it to "predominately marine aerosol", p9, l5.

*• Page 9, Line 2: Similar to the comment above, you can never fully exclude other aerosol sources although they are very unlikely. I would soften the language here.*

The whole paragraph was reshaped.

*• First paragraph on page 10: The discussion on the RH-dependency of the LR could be expanded by relating the presented observations to previous literature (e.g. Ackermann, 1998; Zieger et al., 2011).*

**It is not easy to expand the discussion, as the numerical study by Ackermann does not model cubic particles but only spheres. The study by Zieger et al., 2011 reports measurements of the lidar ratio, but without assessing the specific aerosol type. Most probably it is a mixture of marine and continental aerosol that was found over Cabauw. The modeling study by Kemppinen et al., 2015a has been included for 1064 nm only (p13, l25-28). Measurements of the lidar ratio in marine environments around 40% RH or lower were not found.**

• *Page 10, line 27: Here, I would make it more clear that you have observed a specific case of the dehydration of particles along the upper branch of the hysteresis curve (i.e. that they have fully deliquesced before they are dehydrated again). In addition, one should keep in mind that inorganic sea salt has multiple efflorescence points due to its complex composition (see e.g. Tang et al., 1997; Zieger et al., 2017).*

**Thank you for your comment. The information and the references have been added on p10, l29-31.**

• *Equation 2 and Fig. 13: The upper and lower branches of the hysteresis curve would have to be fitted separately. Therefore, one would not expect a value of A = 1 but rather a value of A > 1 for sea salt aerosol, especially if you consider that most of the particles will have fully deliquesced at one point and be on the upper branch of the hysteresis curve. Since the values of A are here clearly below 1, I assume that the reference RH of the backscattering coefficient is not sufficiently low (or just not known).*

**Thank you for pointing it out. A corresponding discussion about A has been added p12, l3-5.**

• *Equation 2 and Fig. 13: Here, you should also discuss that these results are only valid if the reference RH of 40 − 50% is sufficient to represent 'dry' particles, which, as discussed above, is probably not the case. This will be a critical point for modelers who will relate their 'wet' values of aerosol optical properties to perfectly dry values (at RH=0%). As such, they will subsequently estimate much higher enhancement factors than shown in Table 2 or Fig. 13. Figure 1 below shows the same gamma-fit as used in the presented study. Extrapolating to RH=0% will give a value of approx. A ≈ 1.5 (with gamma = 0.45).*

**Titos et al., 2016 mention that f(RH) will be underestimated by approximately 25%, if the dry reference value is <40% RH. A sentence concerning this issue has been added on p12, l10-11. This gives a rough estimate about how much higher the scattering enhancement factor should be.**

**Thank you for your comments. It would be of great benefit to bring the in situ, laboratory and remote sensing measurements together to better understand the atmosphere and improve the atmospheric models.**

[revised manuscript text omitted]